# Rethinking Transformer for Long Contextual Histopathology Whole Slide Image Analysis

**Honglin Li**[1,3] **Yunlong Zhang**[1,3] **Pingyi Chen**[1,3] **Zhongyi Shui**[1,3]
**Chenglu Zhu**[2,3*] **Lin Yang**[2,3*]
[1] Zhejiang University
[2] Research Center for Industries of the Future and [3] School of Engineering, Westlake University
{lihonglin,zhuchenglu,yanglin}@westlake.edu.cn

## Abstract

Histopathology Whole Slide Image (WSI) analysis serves as the gold standard for clinical cancer diagnosis in the daily routines of doctors. To develop computer-aided diagnosis model for histopathology WSIs, previous methods typically employ Multi-Instance Learning to enable slide-level prediction given only slide-level labels. Among these models, vanilla attention mechanisms without pairwise interactions have traditionally been employed but are unable to model contextual information. More recently, self-attention models have been utilized to address this issue. To alleviate the computational complexity of long sequences in large WSIs, methods like HIPT use region-slicing, and TransMIL employs Nyströmformer as an approximation of full self-attention. Both approaches suffer from suboptimal performance due to the loss of key information. Moreover, their use of absolute positional embedding struggles to effectively handle long contextual dependencies in shape-varying WSIs. In this paper, we first analyze how the low-rank nature of the long-sequence attention matrix constrains the representation ability of WSI modelling. Then, we demonstrate that the rank of attention matrix can be improved by focusing on local interactions via a local attention mask. Our analysis shows that the local mask aligns with the attention patterns in the lower layers of the Transformer. Furthermore, the local attention mask can be implemented during chunked attention calculation, reducing the quadratic computational complexity to linear with a small local bandwidth. Additionally, this locality helps the model generalize to unseen or under-fitted positions more easily. Building on this, we propose a local-global hybrid Transformer for both computational acceleration and local-global information interactions modelling. Our method, Long-contextual MIL (LongMIL), is evaluated through extensive experiments on various WSI tasks to validate its superiority in: 1) overall performance, 2) memory usage and speed, and 3) extrapolation ability compared to previous methods. Our code will be available at https://github.com/invoker-LL/Long-MIL.

## 1 Introduction

Though digital pathology images have been widely used for Cancer diagnosis [50, 65, 89, 94, 17, 44] and prognosis [9, 12, 71] and gene expression [82] via automatic computer-assisted analysis, the Giga-pixels of resolution, as large as $150,000 \times 150,000$ pixels [50, 12, 60] of Whole Slide Image (WSI), still poses great challenges on both annotation labelling and efficient computation for model training [42]. Thus, previous methods [9, 12, 42, 5, 40, 92, 93, 41] focus on developing annotation-

---

*Corresponding Author

38th Conference on Neural Information Processing Systems (NeurIPS 2024).

& computational- efficient learning to cope with those problems by employing Multiple Instance Learning (MIL) [51, 32] with only WSI-level supervision.

Currently, there are mainly three steps (or mainstream genres) of WSI analysis framework: 1) access better instance-level patch embedding via Self-supervised Learning [30, 7, 9, 40, 77].

2) design WSI head architectures [50, 65, 89] and train the head with frozen instance embedding. 3) fine-tune patch embedding with WSI level weak label for better task-specific results [90, 42]. Here in this paper, we focus on the step-2 and uncovering that there are still some room for improvement: Firstly, the vanilla attention used in AB-MIL, DS-MIL, CLAM, etc. [32, 40, 50, 89], despite its computational efficiency (compared to self-attention), is unable to model contextual or interaction information across instances within a WSI. These interactions, which play a crucial role in prediction decision-making [11, 65] indeed, can be modelled via self-attention mechanism. However, the long sequence of WSI instances pose $O(n^2)$ computation complexity with self-attention (Fig. 1). Although this complexity can be alleviated by self-attention approximation methods like Nyströmformer [83, 75] used in Trans-MIL [65], this approximation only get sub-optimal

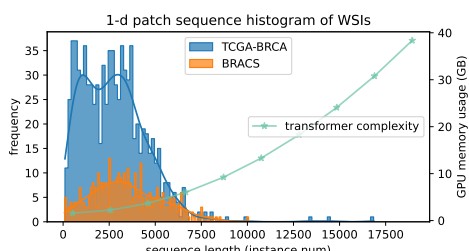

Figure 1: Handling an extremely long sequence with a magnification of $20\times$ (or quadrupling to $40\times$) poses a significant challenge. The computational complexity of transformers, denoted as $O(n^2)$, becomes prohibitive in such cases, leading to computational explosion.

performance compared to self-attention as pointed out in [20, 29]. Authors in HIPT [9] mitigates the complexity by non-overlap large region slicing, but the interactions of instances from different region slicing are highly ignored (e.g. adjacent patches may be separated into two regions).

The above issues highlight a strong need for an effective and efficient Transformer for WSI modelling. To begin, we discuss the performance bottleneck of basic Transformer [72] for WSI. Different to Vision Transformer [23] for natural image modelling where the number of patched tokens are smaller than patch embedding size (e.g. ViT-Small-p16 with embedding size 384 attend on 196+1 tokens), the Transformer-based WSI model suffers severe low-rank bottleneck of attention matrix [3, 22] given the long-sequence ($n >> 1024$) of WSI but limited embedding size ($d \leq 1024$). We reveal this problem in WSI theoretically, thus finding that one self-attention layer with limited embedding size can not model local contexts and global interactions at the same time. By stacking multiple self-attention layers, we notice that the low layer focus more on local context (Fig. 2a) after training while high layer focus on global. However, the rank of the attention matrix is still limited (Fig. 2a+d), resulting in constrained performance.

We assert that the low-rank bottleneck causes the attention mechanism to become confused between local and global interactions, even after training. In other words, using $Q$ and $K^\top$ with only $2dn$ points, it's hard to model $n \times n$ interactions comprehensively in the context of WSI where $n >> d$. We believe that it would be better focusing on less interactions in one layer. Motivated by the low layers of Transformer showing highly sparse attention pattern (Fig. 2a+d) with locality, we propose a local attention mask to learn local interaction more directly. This local mask, more importantly, can highly improve the rank of the attention matrix, showing better representation ability. Furthermore, the local attention mask can be implemented during chunked attention calculation, reducing the quadratic computational complexity to linear with a small local bandwidth. In addition, this locality helps the model generalize to unseen or under-fitted positions more easily (where absolute position embedding used in methods like TransMIL may fail, see Appendix A.2 for more illustration).

Building on this, we propose a local-global hybrid Transformer for both computational acceleration and local-global information modelling.

Our main contributions can be summarized as 3 folds:

1) We firstly theoretically uncover why Transformer model for WSI-MIL analysis fails, based on the low-rank bottleneck of attention matrix for long sequence but limited embedding size. We then further analyze the sparsity and locality pattern of attention matrix empirically to hint our local attention design.

2) We convert the full self-attention into local attention which shows three advantages: higher rank for better representation ability, lighter computational complexity and extrapolation ability for shape-vary WSIs. We further combine the full self-attention for global long-range dependency after stacking layers of local attention.

3) Our WSI-analysis experiments are performed on both diagnosis and prognosis tasks on 4 WSI datasets including Breast, Stomach, Colon and Rectal Carcinoma, which show strong universality of the method and practical potential for real-world applications.

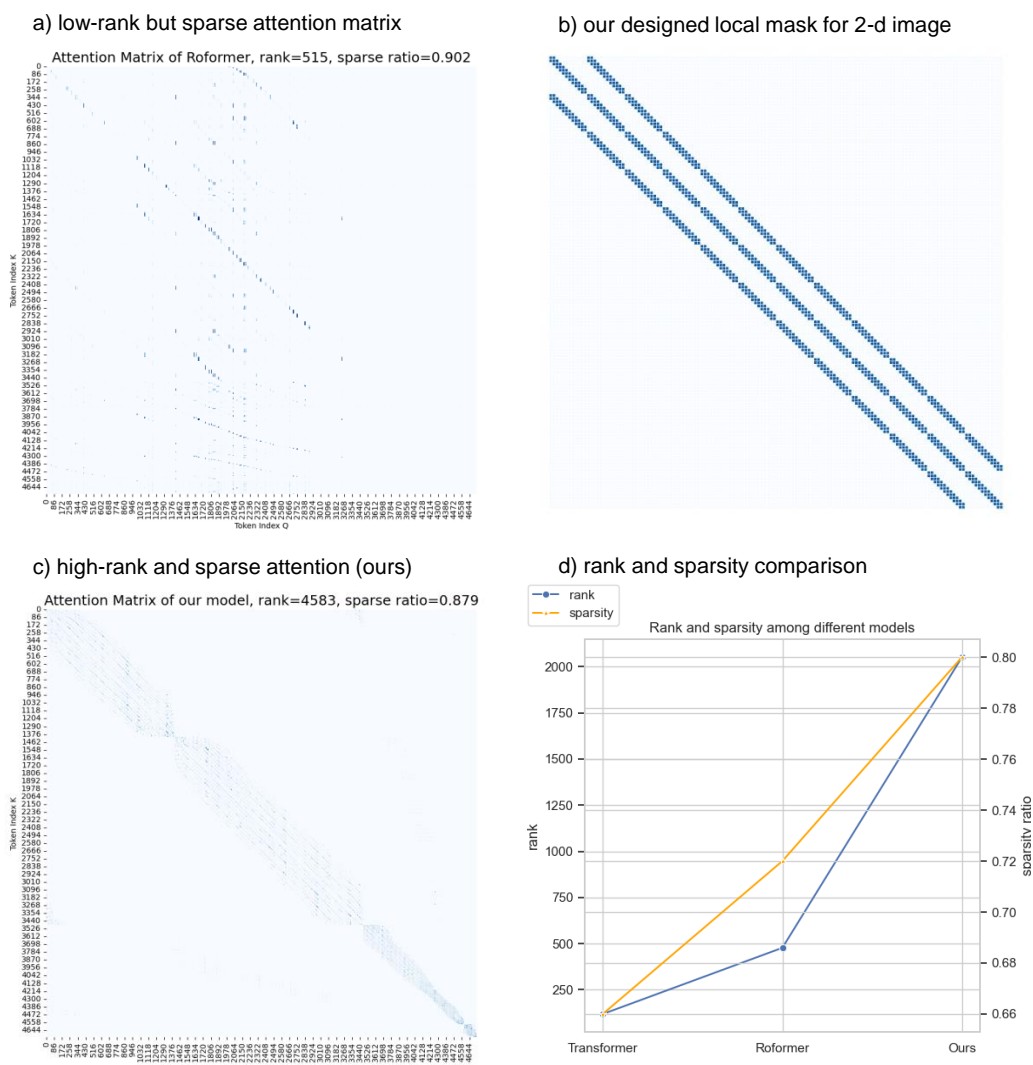

Figure 2: Rank and sparsity of attention matrix in WSI analysis.

## 2 Related Work

### 2.1 Multiple Instance Learning for WSI Analysis

Whole Slide Images (WSIs) contain a rich set of visual information that can aid in pathological analysis [6, 50]. However, accurate annotation of cell-level information within WSIs is labor-intensive and time-consuming [6, 50, 9]. To address this issue, weakly-supervised methods have gained popularity in pathology WSI analysis. Attention-based Multi-Instance Learning (AB-MIL) [32] is adopted to learns instance adaptive weights, allowing the model to focus on informative regions

within the WSIs. This approach significantly reduces the annotation burden of pathologists while still providing valuable insights for patient-level diagnosis. In the context of weakly-supervised pathology WSI analysis, several innovative approaches, DS-MIL, CLAM, DTFD, etc. [32, 40, 50, 89, 35, 61, 18, 2] have been proposed. However, their utilized vanilla attention with light computational cost can not model WSI contextual information, which is useful in pathologist diagnosis decision making [11, 65]. The fine-grained details and global contextual information can also be captured by multi-scale modelling [9, 40]. Graph Network [11, 43, 28, 8, 25] is also useful to make model be context-aware. Similar to this, HIPT [9] and TransMIL [65] have explored the advantages of Transformer with pairwise interactions to model this contextual information. Since Transformer can be generalized to Graph Network [24], both modelling the pairwise interaction, in this paper we focus more on Transformer and try to adapt it better to fit the shape varying and long context properties of WSI. Unlike the authors in [80] who focus on the low-rank properties of pathology images, we investigate from the perspective of low-rank in the attention matrix of Transformer.

## 2.2 Efficient Transformer for Long Sequence

The primary goal of this area is to alleviate the computation and memory complexity of self-attention mechanism on long sequence input. A lot of modifications sparsify the attention matrix [59, 15, 1] with some fixed patterns. Extend to this, some work [73, 74, 64] using learnable patterns in a data-driven fashion, e.g. Reformer [38] introduces a hash-based similarity measure to efficiently cluster tokens into chunks. Linformer [75] technique leverage low-rank approximations of the self-attention matrix, decomposing the $N \times N$ matrix to $N \times k$. The kernels also serve as an approximation of the attention matrix, including Performers [37], Linear Transformers [16]. Another popular method of reducing computation cost is to reduce the resolution of the sequence, hence reducing computation cost by a commensurate factor, e.g. Perceiver [33], Swin Transformer [47]. The recent Nyströmformer [83] used in TransMIL [65], can also be seem as kernel-based low-rank approach. Above work mainly focus on a light approximation of self-attention or using sparse attention, which is indeed worse than the full attention [20]. Recent work like FlashAttention [20] and others [62, 34] using chunked computation scheme and IO-aware mechanism to be memory-efficient and gain full ability like self-attention. Another lines of work try to merge RNN and Transformer, e.g. Transformer-XL [19] proposed a segment-level recurrence mechanism that connects multiple segments and blocks, and now is widely used in most successful LLMs [52, 91, 69]. Recently, linear RNNs [88, 54, 53, 27] and its variants [21, 57] are also proposed, but these recurrent ability is designed for 1-d sequence with causal or auto-regressive property, not fit well for image recognition. To fit longer sequence, better positional embeddings like RoPE, ALiBi, etc. [66, 58, 14] are also proposed. Different to these work focus on NLP task, here in this paper we try to build an efficient Transformer for WSI analysis, which is a unique challenge in vision task.

## 3 Method

### 3.1 Preliminary: Attention-based WSI Analysis

Given a WSI $X$ as input, the goal is to make slide-level prediction $\hat{Y}$ by learning a classifier $f(X; \theta)$. $X$ is firstly patched into a long sequence of small instances $X = \{x_1, ..., x_n\}$ because of its extremely large resolution, where $n$ is the number of instance. The slide-level supervision $Y$ is given by doctors who consider the latent label $y_i$ of all instance $x_i$. Most previous work [6, 50, 9] try to model this process by a Max-pooling operation, so initially, this annotation process is treated as:

$$Y = \max\{y_1, ..., y_n\}. \tag{1}$$

Since the end-to-end training from raw image input to WSI-level output is infeasible because of large memory cost, conventional approaches convert it into two separate stages: Firstly, convert all small patches into instance embeddings $Z = \{z_1, ..., z_n\}$ by a pre-trained backbone such as ResNet [31] or ViT [79], which refers to general features from public ImageNet, or learned on the related dataset to extract the domain-specific representations [9, 36]. Then, aggregate all patches' features within a slide and producing the slide-level prediction $\hat{Y} = g(Z; \theta)$. In this paper, we mainly focus on the latter one, where $g$ is an vanilla attention function followed by a linear classifier head as:

$$\hat{Y} = \sigma(\sum_{i=1}^{n} a_i z_i), \tag{2}$$

where $a_i$ is attention weights and $\sigma(\cdot)$ is a linear head.

However, above vanilla attention method assigning adaptive weight to each instance to make simple summation or pooling can not model the interactions among different instances. Thus, to handle this problem, Transformer with self-attention is employed in TransMIL [65] and HIPT [9], where the attention sublayer computes the attention scores for the $i$-th query $q_i \in R^{1 \times d}$, $(1 \leq i \leq n)$ in each head, where $d$ is the head dimension. In other words, each instance will compute an attention score list as interactions with all instances. These attention scores are then multiplied by the values to return the output of the attention sub-layer as:

$$o_i = \text{softmax}(q_i K^\top)V, \tag{3}$$

where the $\{Q : q_i, K : k_i, V : v_i\} \in R^{n \times d}$ are obtained through linear transform from the input embedding $Z$, the $\text{softmax}(q_i K^\top)$ is the attention score and $O \in R^{n \times d}$ is the output. Given $O$, which encodes the interactions among instances, we can further use Equation (2) and input $O$ to replace $Z$ for final prediction, mean-pooling and class token in ViT [79] can also be adopted. Note that here we omit dropout, FFN, residual connection and some detailed blocks in Transformer for simplicity.

**Positional Embedding:** Since the operation in Equation (3) is position-agnostic, Transformer [72, 23] try to model contextual interactions by incorporate position information. Absolute positional embedding assigns a positional vector $p_m$ to each position $m$ and adds it to the embedding vector as: $z_i = z_i + p_{m,i}$. In HIPT [9], the absolute positional embedding [23] for 2-d is employed, while TransMIL [65] use convolutions as implicit positional embedding [78] but treat data as 1-d sequence. Relative positional embedding that model the positional difference $m - n$ has become popular. Rotary positional embedding (RoPE) [66] encodes the position with rotations: $f(q_m, m) = R_m q_m$, where $R_m$ is a rotation matrix with angles proportional to $m$. With the rotation's property, the query-key product exhibits a positional difference:

$$f(q_m, m)f(k_n, n)^\top = q_m R_{n-m} k_n^\top. \tag{4}$$

The core idea of RoPE is to insert position $m, n$ signal on $q, k$ and reflect the relative position on the newly attention matrix. Though the RoPE is designed for 1-d language sequence, it can also be extended to 2-d paradigm for application on WSI analysis [56].

**Computational Complexity:** Though above Transformer with self-attention can well model the interactions among instances, its computational cost $O(n^2 d)$ is too heavy for long sequence of WSI due to the interactive attention score calculation (see Appendix A.5.5 for 40x magnification WSI modelling). Previous WSI Transformer SOTA like TransMIL [65] and HIPT [9] relieve this problem with different ways: 1) attention approximation: TransMIL [65] utilizes Nyströmformer [83], a mechanism employs kernel-based low-rank approximation to approximate full self-attention for acceleration. 2) region slicing: HIPT [9] utilizes the locality of image by slicing WSI into $4096 \times 4096$ squares without overlapping. Given the fixed window size, in each square there are fixed $16 * 16 = 256$ patches with shape of $256 \times 256$, thus the computational cost can also be seen as linear complexity.

### 3.2 Low-rank and Sparsity of Attention Matrix for Long-sequence WSI

**Low-rank bottleneck:** Considering all queries in $\{Q : q_i\}$, the Equation (3) can be seen as:

$$O = \text{softmax}(QK^\top)V, \tag{5}$$

where the rank of $QK^\top$ can be derived as:

$$r(Q_{n \times d}(K^\top)_{d \times n}) \leq \min(r(Q_{n \times d}), r(K_{n \times d})) = \min(n, d). \tag{6}$$

In the context of WSI analysis, the patched sequence length $n$ of most WSIs is larger than 1024 (Fig. 1), while the embedding size $d$ of the pre-trained patch encoder is less than 1024 (e.g. 1024 in ResNet-50, 384 in ViT-Small, 768 in ViT-Base). Thus, in Equation 6, we have $d \leq 1024 \leq n$, indicating that the rank of the attention matrix in Transformer-based WSI analysis is constrained by the embedding size $d$:

$$r(QK^\top) \leq \min(n, d) = d. \tag{7}$$

As a result, the representation ability of self-attention is limited by the low-rank bottleneck, thus vanilla Transformer based model in WSI analysis suffers sub-optimal performance. Though the

non-linear softmax operation can change the rank, but we still observe limited rank of $\text{softmax}(QK^\top)$ after training (Fig. 2a+d). This is an extremely different problem compared to ViT modelling, e.g. ViT-Small with embedding size of 384 only need to focus on 196 patch tokens (with image size of 224 and patch size of 16), which can model both local contextual and global interactions simultaneously with full-rank attention matrix. We contend that, under this circumstance, Transformer for WSI modelling may become confused when handling local contextual and global interactions with a single layer.

Under this assumption, it is also more easy to understand the limitation of previous SOTA transformer for WSI: TransMIL [65] with Nyströmformer [83] employs kernel-based low-rank approximation to approximate full self-attention. However, it is worth noting that the approximation may produce lower rank of attention matrix compared to basic self-attention, thus resulting lower performance in TransMIL. Similar problems also happens in other models like softmax-free linear attention [54, 67, 85]. We show a lot of experiment of these linear attention model in Appendix A.5.6.

An intuitive modification to handle the low-rank problem is to set a larger embedding size $d$, but this makes computational complexity $O(n^2 d)$ more severe, let alone most pathology patch pre-trained foundation models [36, 49, 10] carry fixed embedding size. Noting that it is infeasible to fully represent a matrix with a shape of $n \times n$ if $n >> d$ given totally $2nd$ feature points from $Q_{n \times d}$ and $K_{n \times d}$, why not focus the attention to more important interactions? Thus in the contrast, we alleviate the low-rank bottleneck together with the problem of computational cost by focusing on locality, motivated by the sparse and local pattern in low layers of attention (Fig. 2a).

**Sparsity with locality:** Here we define the the selection index for retained sparse attention matrix when the softmax probability is greater than a threshold:

$$I = \text{where}\{\text{softmax}(QK^\top) > \tau\}, \tag{8}$$

where the threshold $\tau$ is normalized by sequence number $n$, e.g. $\tau = 0.0001/n$. Then, the sparsity ratio can be denoted as:

$$r = 1 - \frac{\text{len}(I)}{n^2}. \tag{9}$$

We note that there is an obvious sparsity and local pattern in lower Transformer layers (Fig. 2a) under this protocol. Given the learned locality and sparsity, we introduce to learn local contextual information with a addable local attention mask for $A = \text{softmax}(QK^\top)$ (without loss of generality, here we mainly derive on 1-d sequence for simplicity):

$$\text{mask}_{i,j} = \begin{cases} -inf & \text{if } |i - j| > b, \\ 0 & \text{otherwise}, \end{cases} \tag{10}$$

where $b$ is the local band width. Then the attention matrix is converted as:

$$A_{i,j} = \text{softmax}(QK^\top + \text{mask}) = \begin{cases} 0 & \text{if } |i - j| > b, \\ p & \text{otherwise}, \end{cases} \tag{11}$$

where $0 < p < 1$ is the softmax probability.

We claim that our proposed sparse local attention capture both higher rank and reduced computational complexity, which will work better for WSI modelling:

**1) Higher rank:** It is easy to prove that the band matrix $A$ in Equation 11 is of a lower-bound of rank as $n - b$. Here we give a intuitive verification with a small band matrix:

$$A^{<n=9, b=3>} = \begin{bmatrix} a_{11} & a_{12} & a_{13} & a_{14} & 0 & 0 & 0 & 0 & 0 \\ a_{21} & a_{22} & a_{23} & a_{24} & a_{25} & 0 & 0 & 0 & 0 \\ a_{31} & a_{32} & a_{33} & a_{34} & a_{35} & a_{36} & 0 & 0 & 0 \\ a_{41} & a_{42} & a_{43} & a_{44} & a_{45} & a_{46} & a_{47} & 0 & 0 \\ 0 & a_{52} & a_{53} & a_{54} & a_{55} & a_{56} & a_{57} & a_{58} & 0 \\ 0 & 0 & a_{63} & a_{64} & a_{65} & a_{66} & a_{67} & a_{68} & a_{69} \\ 0 & 0 & 0 & a_{74} & a_{75} & a_{76} & a_{77} & a_{78} & a_{79} \\ 0 & 0 & 0 & 0 & a_{85} & a_{86} & a_{87} & a_{88} & a_{89} \\ 0 & 0 & 0 & 0 & 0 & a_{96} & a_{97} & a_{98} & a_{99} \end{bmatrix}. \tag{12}$$

Then, let's consider the lower-left sub-matrix ranging from $(b+1, 1)$ to $(n, n-b)$:

$$A_{sub} = \begin{bmatrix} a_{(b+1)1} & a_{(b+1)2} & \cdots & a_{(b+1)(n-b)} \\ 0 & a_{(b+2)2} & \cdots & a_{(b+2)(n-b)} \\ \vdots & \vdots & \ddots & \vdots \\ 0 & 0 & \cdots & a_{n(n-b)} \end{bmatrix}, \qquad (13)$$

which is apparently a upper triangular matrix with a full rank of $n - b$. Thus, we have:

$$rank(A) \geq rank(A_{sub}) = n - b. \qquad (14)$$

Since $n > 1024 >> b$ practically, our model with higher rank carry stronger representation ability which can focus more on local contextual information.

**2) Reduced computational complexity:** Given a larger $n$ but fixed small $b$ in Equation 12, there will be a lot of zeros in upper-right and lower-left areas. We note that these zeros can be omitted during attention matrix calculation by a chunking method [20]. Since most operations in $softmax(QK^\top)$ can be skipped, the modified complexity $O(bnd)$ will linearly related to sequence length and band-width, which is heavily reduced compared to $O(n^2 d)$.

**3) Extrapolation ability:** Despite above advantages, here we further show that our model can tackle WSIs with varying input shape better, in other words, extrapolation ability. RoPE need to be well trained or fine-tuned on unseen or seldom seen longer length [45, 13, 81]. Another strategy Attention with Linear Bias (ALiBi) adds pre-defined bias term after the query-key dot-product attention matrix before softmax. For the original 1-d ALiBi [58], the bias is a static, non-learned matrix $softmax(q_i k_j^\top - \rho |i - j|$ computed by the distance between tokens from different positions. We also introduce 2-d ALiBi by 2-d Euclidean distance among token positions, and we find that it shows similar pattern (Appendix A.3) compared to our 2-d local attention but it needs to focus on all instances.

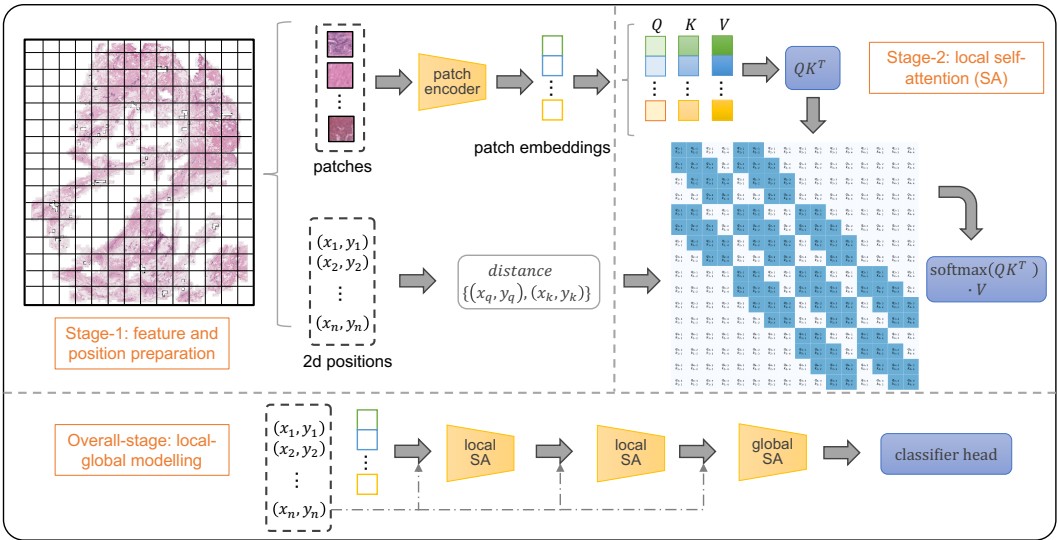

Figure 3: LongMIL framework for WSI local-global spatial contextual information interaction and fusion. 1) Preparing patch feature embedding and 2-d positions of WSIs. 2) Performing pairwise computations among all positions within a WSI by local masking as acceleration. 3) Overall local-global forward of the model, where position information need to be feed to both local (local masking) and global (positional embedding).

### 3.3 LongMIL framework and implementation

To realize long contextual MIL modelling and better WSI analysis performance, the overall framework (as depicted in Fig. 3) of our method includes 3 stages:

1) Segment and patch WSI into instances, then save its corresponding foreground patch feature embedding and 2-d positions for preparation.

2) Calculate the local self-attention matrix by the local window mask given position distance. This process is finished by trunk method like FlashAttention [20] to omit non-masking areas.

3) After multi layers (two as default) of local attention focusing on local contextual information interactions, a pooling function with window size $2 \times 2$ is employed to reduce token number by 4 times. Then a basic self-attention focus on global interactions is computed to get final feature and prediction.

## 4 Experiments

In this section, we present the performance of the proposed method and compare it with various baselines. Ablation experiments are performed to further study the proposed method, for paper length, more experimental results are presented in the Appendix A.5.

**Datasets and Tasks.** We use four datasets to evaluate our method for both **tumor subtyping** and **survival prediction**. For data details and pre-processing, please see Appendix A.4.

Table 1: **Slide-Level Tumor Subtyping** on BRACS by using two pre-trained embeddings. **Top Rows.** Various WSI-MIL architectures with vanilla attention (no interaction among different instances). **Bottom Rows.** TransMIL (using Nyströmformer and learnable absolute position embedding), full attention (+RoPE) and our LongMIL.

| | BRACS tumor subtyping | | | |
|---|---|---|---|---|
| | ViT-S Lunit [36] | | ViT-S DINO (our pre-train) | |
| Method | F1 | AUC | F1 | AUC |
| KNN (Mean) | $0.503_{\pm 0.011}$ | $0.691_{\pm 0.007}$ | $0.430_{\pm 0.029}$ | $0.649_{\pm 0.008}$ |
| KNN (Max) | $0.472_{\pm 0.009}$ | $0.771_{\pm 0.018}$ | $0.416_{\pm 0.019}$ | $0.645_{\pm 0.007}$ |
| Mean-pooling | $0.534_{\pm 0.026}$ | $0.741_{\pm 0.017}$ | $0.487_{\pm 0.034}$ | $0.717_{\pm 0.020}$ |
| Max-pooling | $0.649_{\pm 0.032}$ | $0.843_{\pm 0.018}$ | $0.598_{\pm 0.032}$ | $0.818_{\pm 0.006}$ |
| AB-MIL [32] | $0.668_{\pm 0.032}$ | $0.866_{\pm 0.016}$ | $0.621_{\pm 0.048}$ | $0.837_{\pm 0.035}$ |
| DS-MIL [40] | $0.607_{\pm 0.044}$ | $0.824_{\pm 0.028}$ | $0.622_{\pm 0.063}$ | $0.808_{\pm 0.033}$ |
| CLAM-SB [50] | $0.647_{\pm 0.020}$ | $0.836_{\pm 0.021}$ | $0.627_{\pm 0.032}$ | $0.836_{\pm 0.009}$ |
| DTFD-MIL MaxS [89] | $0.597_{\pm 0.025}$ | $\underline{0.874_{\pm 0.026}}$ | $0.521_{\pm 0.059}$ | $0.807_{\pm 0.016}$ |
| DTFD-MIL AFS [89] | $0.608_{\pm 0.083}$ | $0.869_{\pm 0.018}$ | $0.538_{\pm 0.053}$ | $0.824_{\pm 0.011}$ |
| TransMIL [65] | $0.648_{\pm 0.054}$ | $0.835_{\pm 0.031}$ | $0.591_{\pm 0.049}$ | $0.798_{\pm 0.029}$ |
| Full Attention | $\underline{0.689_{\pm 0.036}}$ | $0.870_{\pm 0.010}$ | $\underline{0.648_{\pm 0.028}}$ | $\underline{0.839_{\pm 0.018}}$ |
| LongMIL (ours) | $\mathbf{0.706_{\pm 0.025}}$ | $\mathbf{0.888_{\pm 0.019}}$ | $\mathbf{0.657_{\pm 0.026}}$ | $\mathbf{0.848_{\pm 0.004}}$ |

**Pre-training Patch Encoders.** Our work mainly focus on the WSI-head results based on some good pre-trained encoders for histopathology including HIPT [9], Lunit [36] and newly foundation models like UNI [10] and GigaPath [84]. We also include ResNet-50 pretrained in ImangeNet-1k and ViT-small pretrained in BRACS patch data by ourself with DINO [7].

**Implementation Details.** We train our model with PyTorch on a RTX-3090 GPU, with a WSI-level batchsize of 1, learning rate of 1e-4, and weight decay of 1e-2. We add positional encoding into the framework, please check our code for details.

### 4.1 Slide-level Tumor Subtyping

**Evaluation Metrics.** For all the experiments, we report the macro-AUC and macro-F1 scores since all these dataset suffering class imbalance. For TCGA-BRCA, we perform 10-fold cross-validation with the same data split adopted in HIPT [9]. Besides, the dataset BRACS is officially split into training, validation and testing, thus the experiment is conducted 5-times with different random seeds. The mean and standard variance values of performance metrics are reported for multi-runs or cross-validation runs.

**Baselines for Comparison.** We first show the results of Mean-/Max- pooling and KNN for traditional evaluation. Then we directly evaluate several classical WSI-MIL methods, including AB-MIL [32],

DS-MIL [40], CLAM [50], DTFD-MIL [89]. Then we compare our method with Full Attention (RoFormer) and TransMIL [65]. We omit HIPT [9] for BRACS since it need WSI larger than a threshold and should based on their pre-trained backbone.

**Results Analysis:** For BRACS 3-categories tumor subtyping, the results are reported in Table 1. We can first observe that both Full Attention and our LongMIL show improvement respectively. For Full Attention, attributing to its full self-attention for pairwise interaction ability, it shows better performance compared to all vanilla attention modules [32, 40, 50] and especially TransMIL [65] which use attention approximation, but it is not quite stable to beat DTFD [89].

For TCGA-BRCA 2-categories tumor subtyping, we show the results in the Appendix A.5.1.

## 4.2 Slide-level Survival Prediction

Table 2: **Slide-Level Survival Prediction** based on HIPT [9] pre-trained embedding with various WSI-MIL architectures including vanilla attention, GCN, TransMIL, self-attention (HIPT with region slicing and absolute embedding), full self-attention and our LongMIL.

| Method | COADREAD | STAD | BRCA |
|---|---|---|---|
| AB-MIL [32] | $0.566_{\pm0.075}$ | $0.562_{\pm0.049}$ | $0.549_{\pm0.057}$ |
| AMISL [86] | $0.561_{\pm0.088}$ | $0.563_{\pm0.067}$ | $0.545_{\pm0.071}$ |
| DS-MIL [40] | $0.470_{\pm0.053}$ | $0.546_{\pm0.047}$ | $0.548_{\pm0.058}$ |
| GCN-MIL [43] | $0.538_{\pm0.049}$ | $0.513_{\pm0.069}$ | - |
| HIPT [9] | $\underline{0.608}_{\pm0.088}$ | $0.570_{\pm0.081}$ | - |
| TransMIL [65] | $0.597_{\pm0.134}$ | $0.564_{\pm0.080}$ | $0.587_{\pm0.063}$ |
| Full Attention | $0.603_{\pm0.048}$ | $\underline{0.568}_{\pm0.074}$ | $\underline{0.601}_{\pm0.047}$ |
| LongMIL (ours) | $\mathbf{0.624}_{\pm\mathbf{0.057}}$ | $\mathbf{0.589}_{\pm\mathbf{0.066}}$ | $\mathbf{0.619}_{\pm\mathbf{0.053}}$ |

**Evaluation Metrics.** For all the experiments, C-Index scores are reported for the 3 datasets. We follow the data splits and pre-trained patch embedding provided in HIPT [9] for fair comparison. The performance results are also reported via the mean and standard variance values of performance metrics by multiple folder cross-validation with the same running setting to HIPT [9].

**Baselines for Comparison.** For this task, we use the survival cross-entropy loss proposed by Zadeh et al. [87]. The results are summarized in Table 2, where we directly evaluate several survival prediction WSI-MIL methods, including AB-MIL [32], AMISL [86], DS-MIL [40], GCN-MIL [43]. Then we compare our method with some state-of-the-art combining position embedding on Transformer: TransMIL [65] and HIPT [9]. Though our method show some improvement, the C-index score is still too low to daily clinical usage depending on only WSI information. In the near future, we would like to investigate more on this task, e.g. combining multi-modality features as used in [12], since Transformer also born with great ability on multi-modality fusion [39, 70, 12, 63].

## 4.3 Evaluation on Pathology Foundation Models

Since recent Pathology Foundation Model (PathFMs) [10, 48, 84] have been emerging as strong patch encoders, we here further provide evaluations based on PathFMS including UNI [10] and GigaPath [84]. The pre-processing procedure is the same to previous sections. Since the WSI params are pre-trained in GigaPath, we also experiment it using random initialization for fair comparison. For the mismatch of UNI patch encoder and GigaPath WSI head, we add a nn.Linear layer as a feature projector. The results is shown in Table 3, we find that our method also show consistency improvement with PathFMs. Furthermore, we find that the pre-training plays a key role to the success of Prov-GigaPath WSI-head, since transformers are much more over-parameterized than previous simple attention-based MIL. However, the WSI-level pretrained model relies on patch encoder (e.g. UNI patch encoder + GigaPath WSI do not show competing result). We also provide more difference analysis of efficient attention mechanism compared to GigaPath WSI head in Appendix A.6. In table 4, we also include survival prediction based on PathFMs.

Table 3: Slide-Level Tumor Subtyping on BRACS based on Pathology Visual Foundation Models.

| Method | BRACS tumor subtyping | | | |
| | UNI [10] | | GigaPath [84] | |
| | F1 | AUC | F1 | AUC |
|---|---|---|---|---|
| AB-MIL [32] | $0.692_{\pm 0.033}$ | $0.875_{\pm 0.020}$ | $0.640_{\pm 0.022}$ | $0.837_{\pm 0.010}$ |
| CLAM-SB [50] | $0.640_{\pm 0.057}$ | $0.844_{\pm 0.025}$ | $0.624_{\pm 0.023}$ | $0.826_{\pm 0.014}$ |
| DTFD-MIL [89] | $0.655_{\pm 0.031}$ | $0.878_{\pm 0.022}$ | $0.610_{\pm 0.032}$ | $0.843_{\pm 0.017}$ |
| TransMIL [65] | $0.592_{\pm 0.036}$ | $0.859_{\pm 0.023}$ | $0.599_{\pm 0.058}$ | $0.838_{\pm 0.048}$ |
| Full Attention | $\underline{0.715_{\pm 0.043}}$ | $\underline{0.884_{\pm 0.017}}$ | $0.663_{\pm 0.023}$ | $0.850_{\pm 0.018}$ |
| GigaPath-random init | $0.648_{\pm 0.041}$ | $0.837_{\pm 0.033}$ | $0.627_{\pm 0.038}$ | $0.808_{\pm 0.038}$ |
| GigaPath-pretrained | $0.668_{\pm 0.026}$ | $0.861_{\pm 0.030}$ | $\mathbf{0.677_{\pm 0.033}}$ | $\mathbf{0.862_{\pm 0.034}}$ |
| LongMIL (ours) | $\mathbf{0.728_{\pm 0.045}}$ | $\mathbf{0.887_{\pm 0.008}}$ | $\underline{0.673_{\pm 0.023}}$ | $\underline{0.856_{\pm 0.015}}$ |

Table 4: TCGA-BRCA Survival Prediction based on Pathology Visual Foundation Models.

| Method | UNI [10] | GigaPath [84] |
|---|---|---|
| AB-MIL [32] | $0.630_{\pm 0.054}$ | $0.635_{\pm 0.033}$ |
| AMISL [86] | $0.627_{\pm 0.080}$ | $0.620_{\pm 0.040}$ |
| DS-MIL [40] | $0.616_{\pm 0.034}$ | $0.612_{\pm 0.086}$ |
| TransMIL [65] | $0.598_{\pm 0.059}$ | $0.599_{\pm 0.064}$ |
| Full Attention | $\underline{0.638_{\pm 0.056}}$ | $\underline{0.617_{\pm 0.069}}$ |
| LongMIL (ours) | $\mathbf{0.656_{\pm 0.061}}$ | $\mathbf{0.645_{\pm 0.055}}$ |

## 4.4 Further Experiments and Ablations

We also provide abundant ablations in Appendix A.5.3 to select best setting including: Transformer blocks and multi-head number, dropout ratio, weight decay, learning rate and the local window size. For linear attention result, please check Appendix A.5.6 with main findings that the linear attentions show low performance like TransMIL since it get low-rank problem more easily. For linear RNN method like Mamba, the result is also relatively lower than Transformer since 2-d WSIs do not hold causal property like 1-d data. For extrapolation validation, please check Appendix A.2 where our method show significant performance improvement (p-value $\approx 0.1$). We also find that our method show consistency improvement when equipped on different magnification (e.g. 40x) and patch size (224 -> 448) as shown in Appendix A.5.4.

## 5 Conclusions and Limitations

In conclusion, our work introduces advancements in computer-aided diagnosis for histopathology WSI analysis. By analyze the low-rank bottleneck and sparsity property and proposing a local-global hybrid Transformer model, our method, Long-contextual MIL (LongMIL), demonstrates superior performance in handling large and shape-varying WSIs. The evaluations across various tasks highlight its accuracy, extrapolation ability, and efficiency compared to previous methods. Our contributions enhance WSI analysis and provide valuable insights for future research. The LongMIL has two limitations: First, its application is restricted to very large image via Transformer modelling. Second, limited embedding size is adopted for practical aim and fair comparison, which is the key to stronger performance based on the low-rank assumption. Future work will aim to address these limitations.

## 6 Acknowledgements

This study was partially supported by Zhejiang Provincial Natural Science Foundation of China (Grant no. XHD23F0201), the National Natural Science Foundation of China (Grant no. 92270108), and the Research Center for Industries of the Future (RCIF) at Westlake University.

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

# A   Appendix

## A.1   Why linear complexity is important

We find that FlashAttention [20] using memory-efficient trucking and hardware IO-aware operations is good enough in both memory and speed to cope with 20x magnification (about twice slower) WSI model (as a result we use Full Attention as the last global attention layer of our hybrid Long-MIL model for 20x magnification, we also provide a replacement version using linear attention as last layer as shown in Table 9 signed as LongMIL+V-Mamba). However it is unacceptable in dealing with 40x magnification (about 30-times slower in BRACS), which takes us 2 days or more to train 5-fold runs models on BRACS, and it takes longer if stacking more layers and on larger WSI with more rounds (e.g.average over 50000 instances of TCGA-BRCA dataset with double WSI num and traditionally using 10 fold-cross validation). This hinders the improvement of 40x magnification ( which includes more useful details) for both development and deployment. Thus we use LongMIL+V-Mamba (hybrid transformer as local+local+linear global attention) in 40x and also get a strong performance as shown in Table 7 left-column, which is faster in speed and comparable in performance compared to FlashAttention.

## A.2   Extrapolation: Train small but test large

We first split BRACS dataset into training (small images) and validation+testing (large image) by sorting them via instance number and then use train-val-test ratio as 6:2:2. The experimental results are plot in lower-right area of Fig. 4.

## A.3   HIPT Region Slicing, Local-mask Matrix and 2-d ALiBi

In Fig. 5, we show the difference and similarity between HIPT region slicing, local-mask matrix and 2-d ALiBi, where our local mask can be seen as a generalization of HIPT and 2-d ALiBi.

## A.4   Details of Datasets

For the slide-level **tumor subtyping** performance, our method is evaluated on two datasets:

BReAst Carcinoma Subtyping (BRACS) [4] collect H&E stained Histology Images, containing 547 WSIs for three lesion types, i.e., benign, malignant and atypical, which are further subtyped into seven categories. Here, since the WSIs number is limited, we only perform three class subtyping. The WSIs are segmented in $20\times$ magnification and non-overlapping patching with $224 \times 224$ size. The Cancer Genome Atlas Breast Cancer (TCGA-BRCA) [68, 55] is a public dataset for breast invasive carcinoma cohort for Invasive Ductal Carcinoma versus Invasive Lobular Carcinoma subtyping. The WSIs are segmented into non-overlapping tissue-containing patches with a size of $256 \times 256$ (keep consistency to previous work [9]) at $20\times$ magnification patches were curated from 1038 WSIs. For the slide-level **survival prediction**, despite TCGA-BRCA, we further includes 2 TCGA histology datasets: 1) A combination dataset of the Colon adenocarcinoma and Rectum adenocarcinoma Esophageal carcinoma (TCGA-COADREAD), which includes 316 WSIs as used in HIPT [9]. 2) Stomach adenocarcinoma (TCGA-STAD) dataset including 321 WSIs. For pre-processing, we using the implementation of CLAM [50] which mainly includes HSV, Blur, Thresholding, and Contours methods to localize the tissue regions in each WSI.

## A.5   Further Experiments and Ablations

### A.5.1   TCGA-BRCA 2-categories tumor subtyping

The results are reported in Table 5 and right column of Table 6. We could observe a significant improvement in our method when using HIPT pre-trained patch embeddings, but only a slight improvement with Lunit. This could be because this task reaches an upper bound with the high-quality Lunit embeddings. Given that it only predicts binary categories, even simple max-pooling can outperform almost all previous MIL methods.

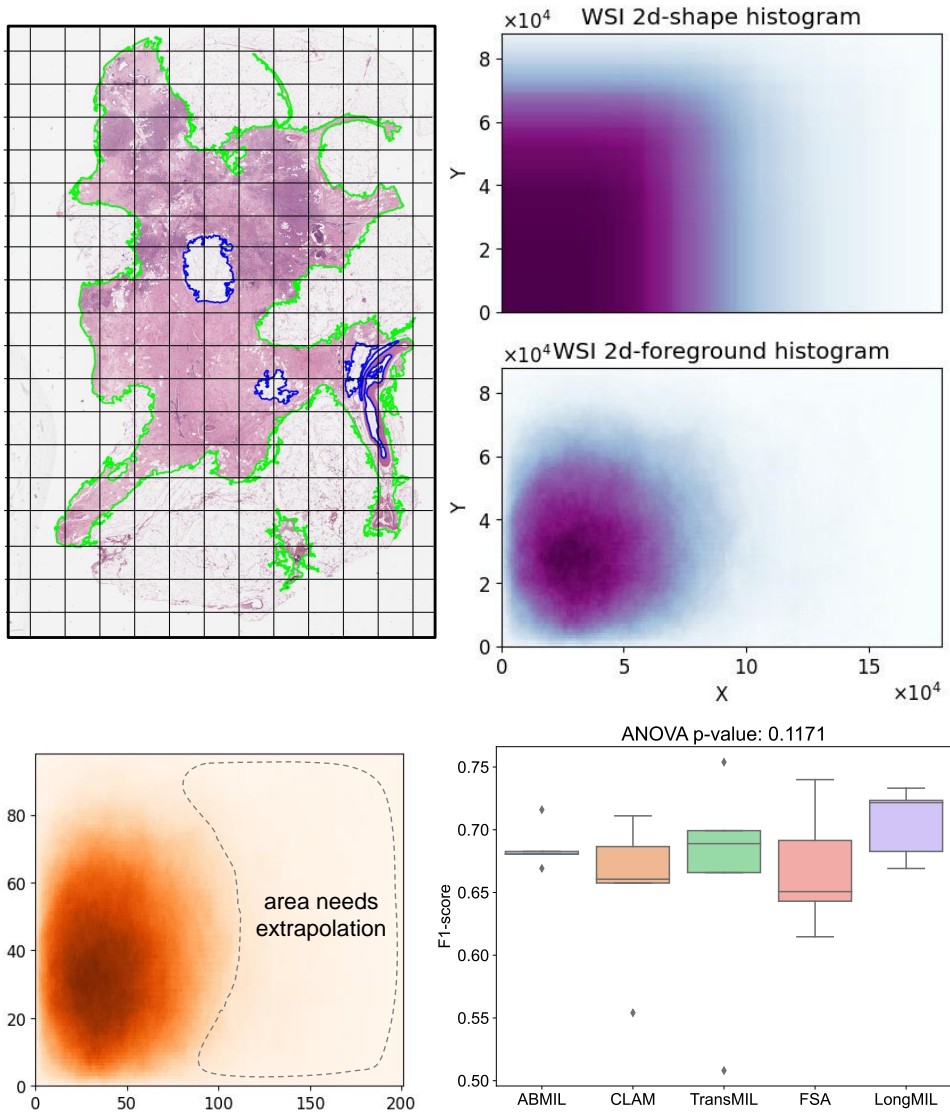

Figure 4: **upper left:** The WSI fore-ground shows irregularity (inner the green line). **upper right and lower left:** The 2-d position index of WSI foreground patches mainly scattered within index<100, thus area enclosed by the dashedline suffers under-fitting with previous method. **lower right:** TransMIL and full self-attention (FSA) get a relatively low performance during testing on unseen larger WSI. Assisted by our method, this case show significant performance improvement (p-value near 0.1).

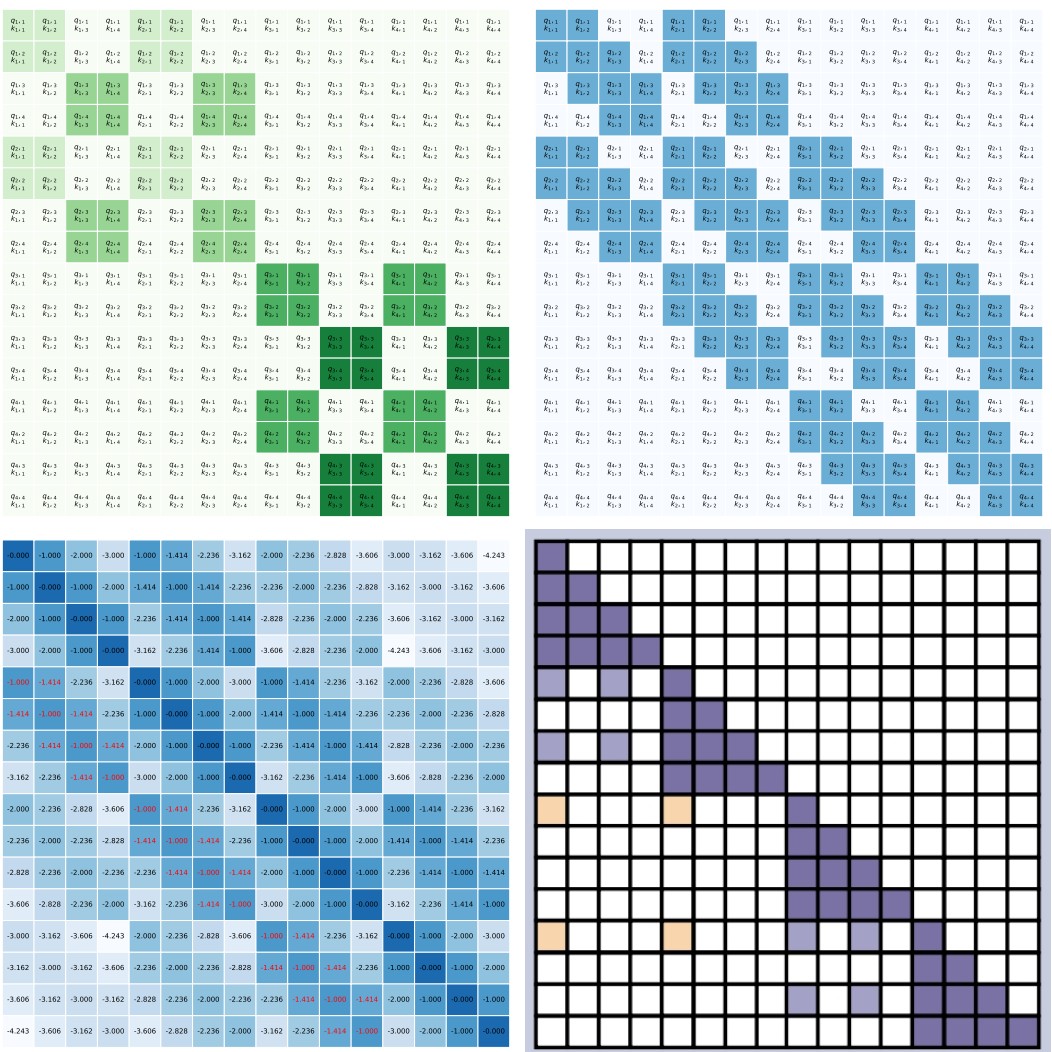

Figure 5: Difference and similarity between various methods. **upper left:** HIPT slicing with extremely hard pattern, **upper right:** our proposed local mask, **lower left:** 2-d ALiBi, or 2-d Euclid distance, **lower right:** attention mask of Prov-GigaPath from their paper (their causal attention, only focus on lower triangular matrix, may be a drawing problem). Apparently the local mask of Prov-GigaPath mainly focus on 1-d interactions (weigh x-axis of WSI more than y-axis), e.g. the interactions when distance less than 2.0 are almost missed, as depicted in the red text areas of the lower-left 2-d Euclid distance subfigure. We have checked their code implementation, which directly apply 1-d LongNet to the serialized (via z-scan) patch sequence.

| Method | TCGA-BRCA tumor subtyping | | | |
| | ViT-S Lunit [36] | | ViT-S HIPT [9] | |
| | F1 | AUC | F1 | AUC |
|---|---|---|---|---|
| KNN (Mean) | $0.669_{\pm 0.088}$ | $0.821_{\pm 0.038}$ | $0.585_{\pm 0.048}$ | $0.742_{\pm 0.016}$ |
| KNN (Max) | $0.657_{\pm 0.069}$ | $0.799_{\pm 0.036}$ | $0.516_{\pm 0.033}$ | $0.691_{\pm 0.016}$ |
| Mean-pooling | $0.841_{\pm 0.050}$ | $0.934_{\pm 0.024}$ | $0.731_{\pm 0.049}$ | $0.867_{\pm 0.037}$ |
| Max-pooling | $\underline{0.849}_{\pm 0.051}$ | $\underline{0.949}_{\pm 0.022}$ | $0.688_{\pm 0.074}$ | $0.826_{\pm 0.058}$ |
| AB-MIL [32] | $0.820_{\pm 0.037}$ | $0.928_{\pm 0.023}$ | $0.757_{\pm 0.069}$ | $0.873_{\pm 0.036}$ |
| DS-MIL [40] | $0.841_{\pm 0.047}$ | $0.925_{\pm 0.024}$ | $0.723_{\pm 0.068}$ | $0.854_{\pm 0.036}$ |
| CLAM-SB [50] | $\mathbf{0.850}_{\pm \mathbf{0.039}}$ | $0.942_{\pm 0.020}$ | $0.733_{\pm 0.057}$ | $0.861_{\pm 0.041}$ |
| DTFD-MIL MaxS [89] | $0.812_{\pm 0.044}$ | $0.911_{\pm 0.031}$ | $0.678_{\pm 0.082}$ | $0.781_{\pm 0.067}$ |
| DTFD-MIL AFS [89] | $0.843_{\pm 0.035}$ | $0.931_{\pm 0.015}$ | $0.704_{\pm 0.075}$ | $0.851_{\pm 0.056}$ |
| TransMIL [65] | $0.824_{\pm 0.026}$ | $0.933_{\pm 0.019}$ | $0.715_{\pm 0.061}$ | $0.840_{\pm 0.053}$ |
| HIPT [9] | - | - | $0.752_{\pm 0.042}$ | $\underline{0.874}_{\pm 0.060}$ |
| Full Attention | $0.843_{\pm 0.060}$ | $0.944_{\pm 0.024}$ | $\underline{0.758}_{\pm 0.046}$ | $0.852_{\pm 0.046}$ |
| LongMIL (ours) | $0.845_{\pm 0.046}$ | $\mathbf{0.950}_{\pm \mathbf{0.023}}$ | $\mathbf{0.762}_{\pm \mathbf{0.064}}$ | $\mathbf{0.880}_{\pm \mathbf{0.045}}$ |

Table 5: Slide-Level Tumor Subtyping on TCGA-BRCA.

### A.5.2 ResNet-50 ImageNet pre-trained embedding results of tumor subtyping

The results experimented in Table 6.

| Method | ResNet-50 ImageNet pre-trained embedding | | | |
| | BRACS | | TCGA-BRCA | |
| | F1 | AUC | F1 | AUC |
|---|---|---|---|---|
| Mean-pooling | $0.483_{\pm 0.018}$ | $0.710_{\pm 0.004}$ | $0.751_{\pm 0.049}$ | $0.861_{\pm 0.026}$ |
| Max-pooling | $0.495_{\pm 0.018}$ | $0.763_{\pm 0.005}$ | $0.780_{\pm 0.027}$ | $0.886_{\pm 0.301}$ |
| AB-MIL [32] | $0.553_{\pm 0.033}$ | $0.752_{\pm 0.005}$ | $0.760_{\pm 0.046}$ | $0.851_{\pm 0.057}$ |
| DS-MIL [40] | $\underline{0.564}_{\pm 0.037}$ | $\underline{0.779}_{\pm 0.032}$ | $\underline{0.797}_{\pm 0.036}$ | $0.894_{\pm 0.029}$ |
| CLAM-SB [50] | $0.548_{\pm 0.026}$ | $0.769_{\pm 0.007}$ | $0.779_{\pm 0.035}$ | $0.878_{\pm 0.027}$ |
| TransMIL [65] | $0.500_{\pm 0.054}$ | $0.734_{\pm 0.019}$ | $0.741_{\pm 0.126}$ | $0.854_{\pm 0.051}$ |
| Full Attention | $0.544_{\pm 0.037}$ | $0.775_{\pm 0.018}$ | $\mathbf{0.800}_{\pm \mathbf{0.014}}$ | $0.901_{\pm 0.014}$ |
| LongMIL (ours) | $\mathbf{0.591}_{\pm \mathbf{0.084}}$ | $\mathbf{0.810}_{\pm \mathbf{0.038}}$ | $0.781_{\pm 0.047}$ | $\mathbf{0.919}_{\pm \mathbf{0.008}}$ |

Table 6: Slide-Level Tumor Subtyping on BRACS and TCGA-BRCA based on ResNet-50 embedding pre-trained via ImageNet supervised learning.

### A.5.3 Hyper-parameters of Transformer training

Number of Transformer blocks and multi-head, bias slope coefficient and local window size, weight decay and dropout ratio: Here we include following hyper-parameters for our results on BRACS with ViT-S patch embedding pre-trained by [36]: Transformer blocks and multi-head number, dropout ratio, weight decay, and learning rate, finally the local window size. Due to time-consumption, we fixed other hyper-parameters when ablation on selected variant (The default setting is Transformer local blocks number = 2, multi-head number = 1, dropout ratio = 0.0, weight decay = 1e-2, and learning rate = 1e-4, local-window size = 10 (radius)), the details can be found in Fig. 6.

### A.5.4 Multi-scale and magnification

There are large differences in speed and performance for 20x and 40x magnification, since FlashAttention [20] will be quite slow if given over 20k instances compared to linear attention our local attention. For performance and speed please check Table 7 and Fig. 7b, respectively.

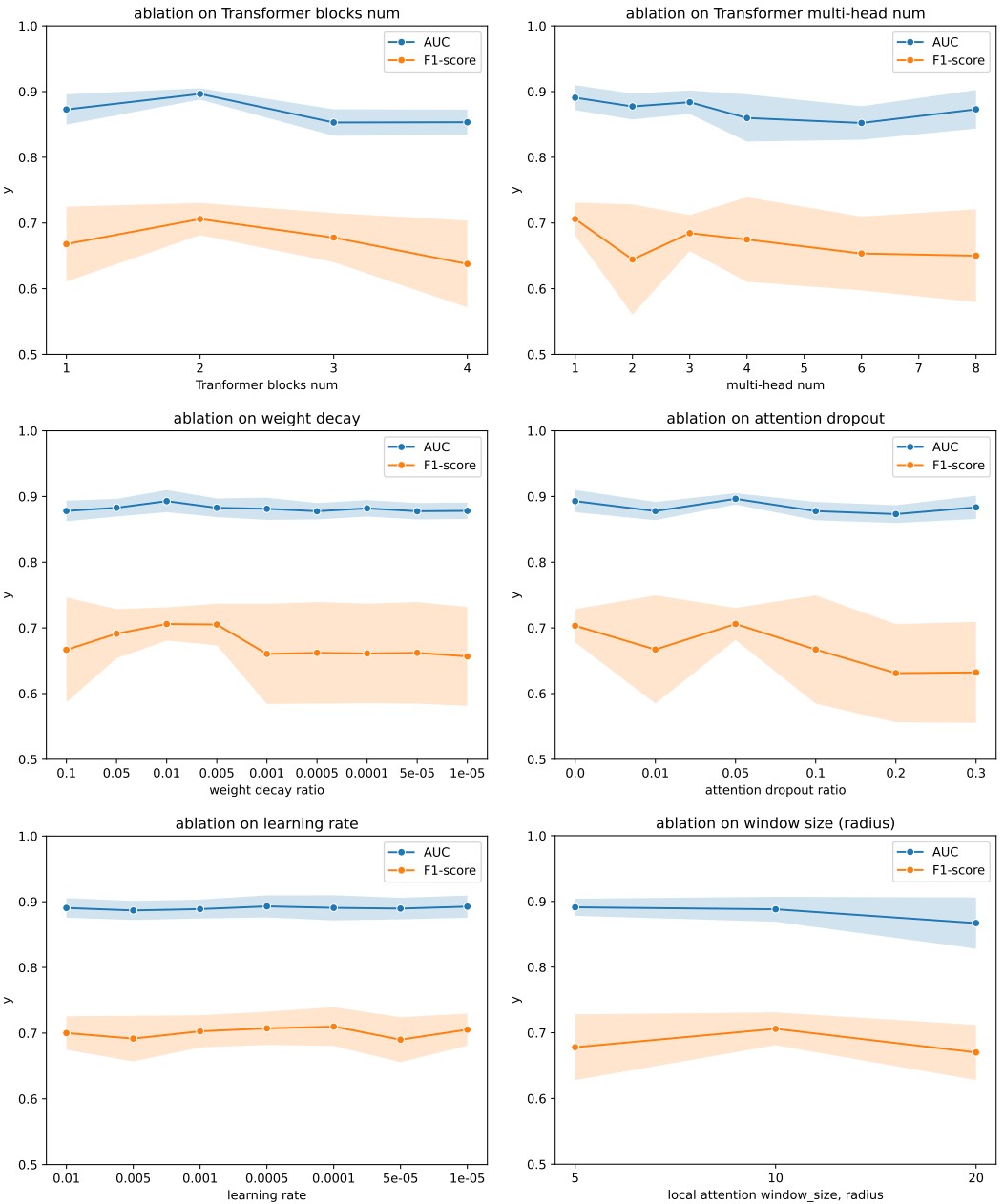

Figure 6: Ablations results on BRACS with ViT-S Lunit [36] patch embedding.

| | BRACS tumor subtyping | | | |
| | 40x | | 20x | |
| Method | F1 | AUC | F1 | AUC |
|---|---|---|---|---|
| AB-MIL [32] | $0.610_{\pm 0.034}$ | $0.811_{\pm 0.013}$ | $0.668_{\pm 0.032}$ | $0.866_{\pm 0.016}$ |
| TransMIL [65] | $0.576_{\pm 0.059}$ | $0.777_{\pm 0.019}$ | $0.648_{\pm 0.054}$ | $0.835_{\pm 0.031}$ |
| Full Attention | $0.618_{\pm 0.042}$ | $0.831_{\pm 0.014}$ | $0.689_{\pm 0.036}$ | $0.870_{\pm 0.010}$ |
| LongMIL (full global) | $\mathbf{0.624_{\pm 0.060}}$ | $\mathbf{0.842_{\pm 0.022}}$ | $\mathbf{0.706_{\pm 0.025}}$ | $\mathbf{0.888_{\pm 0.019}}$ |
| LongMIL (linear global) | $\underline{0.622_{\pm 0.055}}$ | $\underline{0.835_{\pm 0.026}}$ | $\underline{0.693_{\pm 0.024}}$ | $\underline{0.870_{\pm 0.016}}$ |

Table 7: **Ablations on magnification** (40x and 20x) in BRACS tumor subtyping.

We also experiment on larger patch size (from 224 to 448, Table 8) to decrease the overall token number, but find that our method still shows stronger performance.

1. Simple attentions (ABMIL, CLAM without pair wise interactions) gain improvement, and we speculate that the larger image-size can modelling the local context better.

2. DTFD try to split the whole bag into 3 sub-bags, but smaller bag size may result in larger label noise of sub-bags which may answer its performance drop.

3. The gap between LongMIL and TransMIL decreases given closer n and d. Full attention and LongMIL show small drops, since less interactions can be modelled with less patches.

4. LongMIL still out-performs full attention. We speculate that local attention also works better when dealing with the shape-varying WSI even with less n.

| Method | BRACS tumor subtyping | | | |
| | 224 | | 448 | |
| | F1 | AUC | F1 | AUC |
|---|---|---|---|---|
| AB-MIL [32] | $0.692_{\pm 0.03}$ | $0.875_{\pm 0.02}$ | $0.695_{\pm 0.01}$ | $0.875_{\pm 0.01}$ |
| CLAM-SB [50] | $0.640_{\pm 0.06}$ | $0.844_{\pm 0.03}$ | $0.654_{\pm 0.03}$ | $0.851_{\pm 0.02}$ |
| DTFD-MIL [89] | $0.655_{\pm 0.03}$ | $0.878_{\pm 0.02}$ | $0.625_{\pm 0.03}$ | $0.839_{\pm 0.01}$ |
| TransMIL [65] | $0.592_{\pm 0.04}$ | $0.859_{\pm 0.02}$ | $0.646_{\pm 0.07}$ | $0.855_{\pm 0.02}$ |
| Full Attention | $0.715_{\pm 0.04}$ | $0.884_{\pm 0.02}$ | $0.700_{\pm 0.04}$ | $0.874_{\pm 0.02}$ |
| LongMIL | $\mathbf{0.728}_{\pm 0.05}$ | $\mathbf{0.887}_{\pm 0.01}$ | $\mathbf{0.722}_{\pm 0.04}$ | $\mathbf{0.883}_{\pm 0.01}$ |

Table 8: **Ablations on patch size** (224 and 448) in BRACS tumor subtyping based on UNI feature.

### A.5.5 Memory efficiency and speed

We show the memory efficiency and speed of various transformer structures in Fig. 7.

### A.5.6 Linear Attention

We provide ablation on different linear attention e.g. RetNet, GLA [67, 85] and linear RNN structure like Mamba [26] to uncover their advantages and limitations in WSI analysis. As shown in Table 9, we first show the results of these vanilla Linear attention or RNN directly as MIL model (first row), but none of these methods can compete with Full Attention in performance. Then, we combine these modules into our LongMIL to replace its last global attention layer and we observe that this can provide us strong performance as well as linear complexity in total (2 layers of local attention + 1 layer of linear attention, better than two layers of Full Attention in both speed and performance).

### A.6 Detailed comparison to Prov-GigaPath

1. The motivation /contribution: our paper not only focus on proposing an efficient self-attention mechanism for WSI, but also showing analysis on why some previous work like Roformer and TransMIL fail for WSI from the low-rank perspective, which we believe to be insightful to the digital pathology community. However, both the Prov-GigaPath [84] and LongViT [76] focus on scaling up to a large-scale of data with pre-training, which is more empirical. We believe that our analysis may also work for Prov-GigaPath and could be one potential explain on why Prov-GigaPath success and how to improve further.

2. The method details: Prov-GigaPath does not treat interactions inside x-axis and y-axis equally, though the 2-d positional embedding is applied. By putting all patches into a 1-d sequence in a 'z-scan' manner like ViT, their 1-d local attention focus more on x-axis but less on y-axis, as depicted in Fig. 5. Although this can be alleviated by their higher-level dilated attention term, the x-y inequality still exists. Whereas, our local-attention is designed for 2-d (based on 2d Euclid distance), thus treat them equally.

3. The pretrained Prov-GigaPath WSI-head seems relying heavily on their own patch-pretrained encoder, which may be a potential barrier to wide usage, e.g. there are still some cases when GigaPath

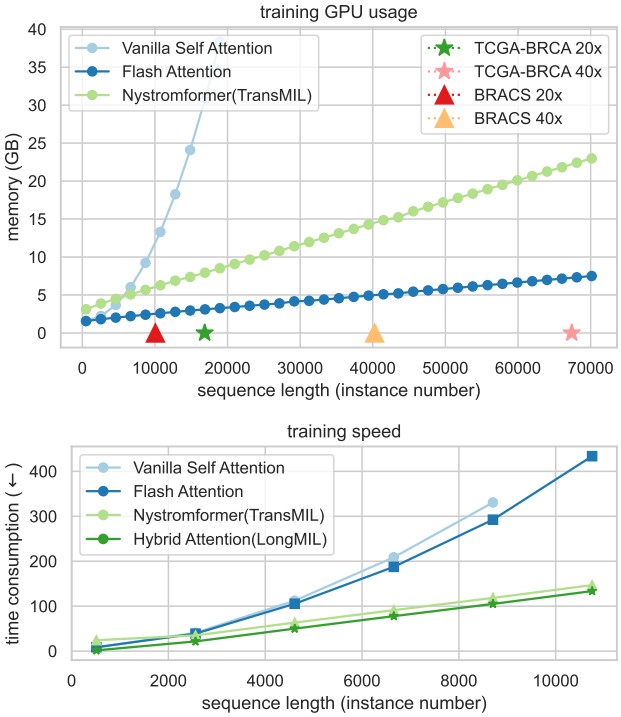

Figure 7: Training memory usage and speed using different Attentions. **Upper**: The chunk method (depicted as Flash Attention) for self-attention calculation convert memory complexity to linear, even better than Nyströmformer. **Lower**: lash Attention with chunk method still suffers quadratic complexity in speed even with hardware-aware accelerated operations. Our introduced LongMIL for WSI analysis can convert it as linearity with local-window mask. We also show markers about the max instance number of WSI used in this paper to show potentials on higher (e.g. 40x) magnification learning. We omit more instances (e.g. over 50k) speed test since it takes a long time, but based on its quadratic complexity of full self-attention, it will be about 25 to 35 times slower than linear attention.

patch features weaker than UNI or Conch, as posted in the github repo of UNI. The WSI pretraining is indeed useful as the key to their superior performance, which covers their problem of spatial inequality on x and y. When dealing with the case 'BRACS', as shown in the following table, our method (even AB-MIL) with better UNI feature can outperform their 'worse patch feature with stronger pretrained slide encoder'.

| Method | BRACS tumor subtyping | |
| --- | --- | --- |
| | F1 | AUC |
| AB-MIL [32] | $0.668_{\pm 0.032}$ | $0.866_{\pm 0.016}$ |
| TransMIL [65] | $0.648_{\pm 0.054}$ | $0.835_{\pm 0.031}$ |
| RetNet [67] | $0.628_{\pm 0.034}$ | $0.805_{\pm 0.009}$ |
| GLA [85] | $0.589_{\pm 0.032}$ | $0.794_{\pm 0.013}$ |
| Mamba [26](random) | $0.650_{\pm 0.024}$ | $0.816_{\pm 0.028}$ |
| Mamba (single) | $0.633_{\pm 0.094}$ | $0.834_{\pm 0.037}$ |
| V-Mamba [46] (cross) | $0.642_{\pm 0.060}$ | $0.821_{\pm 0.028}$ |
| Full Attention | $0.689_{\pm 0.036}$ | $\underline{0.870_{\pm 0.010}}$ |
| LongMIL (ours) | $\mathbf{0.706}_{\pm \mathbf{0.025}}$ | $\mathbf{0.888}_{\pm \mathbf{0.019}}$ |
| + RetNet | $0.690_{\pm 0.051}$ | $0.848_{\pm 0.013}$ |
| + GLA | $0.667_{\pm 0.037}$ | $0.860_{\pm 0.014}$ |
| + Mamba (random) | $0.678_{\pm 0.044}$ | $0.856_{\pm 0.030}$ |
| + Mamba (single) | $0.650_{\pm 0.052}$ | $0.838_{\pm 0.027}$ |
| + V-Mamba | $\underline{0.693_{\pm 0.024}}$ | $\underline{0.870_{\pm 0.016}}$ |

Table 9: **Experiment on Linear Attention** and combine it into our LongMIL as hybrid local-local-linear-attention Transformer model.

