# OpenReview forum: "Rethinking Transformer for Long Contextual Histopathology Whole Slide Image Analysis"
_NeurIPS.cc/2024/Conference — NeurIPS 2024 poster_

### Official Review · Reviewer_UesU · 2024-07-08

**Soundness:** 3
**Presentation:** 2
**Contribution:** 3
**Rating:** 6
**Confidence:** 5

**Summary:**

The authors provide LongMIL, a hierarchical and hybrid of local and global attention mechanism, to address the inherent low-rank bottleneck of MIL problems in computational pathology. Through extensive evaluations across feature encoders and subtyping/survival tasks, the authors indeed demonstrate the superior performance as well as computational efficiency of LongMIL.

**Strengths:**

Despite the recipes for LongMIL being simple (Masking of attention), I find the problem well-motivated and solution intuitive enough to be implemented in future MIL studies. It's been long well-known that TransMIL, although being the first self-attention mechanism in MIL, have not been demonstrating good performance and was in need of some alternative implementations.

I also appreciate the fact that the authors performed extensive ablation studies over several different feature encoder choices as well as tasks of different nature (subtyping and survival), to truly show that LongMIL can be a meaningful contribution to the field.

**Weaknesses:**

There are several weaknesses of the studies that I think the authors need to address for this to be a meaningful contribution to the field.

**Novelty**: Although the authors tried hard to distance from HIPT, I still consider LongMIL solution to be very similar to HIPT - For HIPT, if the first patch-level stage ViT is replaced with the pretrained feature extractors, wouldn't this be same solution as LongMIL, with the difference being how ROI regions are masked? Can authors expand on this point?

**Motivation**: While I agree that the low-rank nature of MIL problem is problematic due to n >> d, I am not entirely convinced that "the representation ability of self-attention is limited by the low-rank bottleneck, thus vanilla Transformer based model in WSI analysis suffers sub-optimal performance" is always the case.

1. There exists lots of morphological redundancy in WSI, so the effective number of distinct features might be much lower than actual n [1], [2]. Therefore low-rank might not always be the issue? Can authors expand on this point?

2. To concretely show n>>d is indeed the issue, the authors should also evaluate their algorithm on tissue biopsies (not tissue resections), where the number of patches would be way lower (few thousand patches) or TMAs (few hundred patches). On these datasets, the gap between LongMIL and other frameworks should decrease, since this is not a low-rank setting.

**Presentation**: Although the paper was not hard to follow, there are several items that needs improvement. There are lots of typos in the paper that need to be ironed out (e.g., line 157 "are got", line 172 "is design", line 298 "which may because"). I am not sure if Figure 2 contributes meaningfully to the paper (also it's impossible to read the axial labels). My suggestion would be to make it smaller and use extra space for the experiments section. Same goes for Equation 12 - I think this can be moved to supplemental section.

The "pre-training backbones" section (line 263~276) was hard to follow. Perhaps make it as a table?

**Experimentation**: To follow up on the Motivation section, I think the authors could run few more ablation studies to demonstrate the severity of low rank issue, by trying to reduce the gap between n and d. This could include larger patch size (256->512) or random sampling patches, both of which have been used in literature and results in lower number of patches.

I think the authors emphasize the survival experimentation over the subtyping (BRACS), since prognosis is known to depend on context [3], [4].

The authors might also consider using latest pathology foundation models (all of these have not been trained on TCGA) - PLIP, UNI, GigaPath - for future studies (It will be too much to do in the given time)

References
[1] Song, Andrew H., et al. "Morphological prototyping for unsupervised slide representation learning in computational pathology." Proceedings of the IEEE/CVF Conference on Computer Vision and Pattern Recognition. 2024.

[2] Vu, Quoc Dang, et al. "Handcrafted Histological Transformer (H2T): Unsupervised representation of whole slide images." Medical image analysis 85 (2023): 102743.

[3] Lee, Yongju, et al. "Derivation of prognostic contextual histopathological features from whole-slide images of tumours via graph deep learning." Nature Biomedical Engineering (2022): 1-15.

[4] Jaume, Guillaume, Andrew H. Song, and Faisal Mahmood. "Integrating context for superior cancer prognosis." Nature Biomedical Engineering 6.12 (2022): 1323-1325.

**Questions:**

- Since n >> d is MIL model-agnostic, would other approaches such as AB-MIL also suffer from low rank nature?
- It seems LongMIL supersedes existing positional encodings (Or am I undsteranding this correctly?) - Can they be combined?
- Equation 11, p -> p_{i,j}. Otherwise, readers might confuse it as being uniform probability.

**Limitations:**

Please see Weakness & Questions.

---

> ### Author Rebuttal · Authors · 2024-08-06
>
> Dear Reviewer UesU,
>
> We appreciate your time and valuable feedback. We are glad that you found that our method is well-motivated and intuitive. Below, please find our point-to-point response to your comments:
>
> > **W1: Novelty: Although the authors tried hard to distance from HIPT, I still consider LongMIL solution to be very similar to HIPT ...**
>
> For the comparison to HIPT, please find the A.4 and Figure 6 of paper manuscript for illustration. Here we provide some details:
>
> * HIPT first slice the whole image into regions (4096x4096, ~50 regions per WSI), then each region *r_4096* is sliced into patches (256x256, 256 patches per region). Each patch *p_256* are extracted by a ViT as feature, the patch-level operation is same as ours and other methods.
> * However, their self-attention on extracted patch features only focuses inside on each region *r_4096* with a Transformer layer. As pointed out in our paper (line 56-57, 180-183), **the adjacent patches may be separated into two regions, and the interactions between them are ignored in HIPT**. The features of each region *r_4096* is processed by further pooling, and a higher Transformer layer globally focus on slide-level.
> * Conversely, **our method do not need 4096 region slicing but use 2-d attention mask to treat all patches equally, thus all patches will interact with other adjacent patches**.
>
> > **W2.1: Motivation: .... There exists lots of morphological redundancy in WSI, so the effective number of distinct features might be much lower than actual n. Therefore low-rank might not always be the issue ...**
>
> We find this question being very constructive to our paper. Though in patch-level the morphology and feature semantic is quite similar, in region-level, similar patches with different spatial combinations may construct different tumor type (carcinoma in situ V.S. invasive cancer). Some previous methods, e.g. DSMIL [1] [2] try multi-scale information to solve this problem, which indicates the importance. Our method can reach similar goal as multi-scale by adaptively local-global interactions in a uniformed transformer, which we believe to be more elegant.
>
> >**W2.2 + W4.1: To concretely show n>>d is indeed the issue... evaluate their algorithm on tissue biopsies (not tissue resections), where the number of patches would be way lower ... On these datasets, the gap between LongMIL and other frameworks should decrease, since this is not a low-rank setting. .... the authors could run few more ablation studies to demonstrate the severity of low rank issue, by trying to reduce the gap between n and d ...***
>
> We run some experiments on larger patch size 448 on BRACS (224 originally), where the largest patch num n is less than 2k, near the feature size of UNI (given the limited time, we will further run biopsies and TMAs in next version). The results are shown in the following table, with main findings that:
>
>     1. Simple attentions (ABMIL, CLAM without pair wise interactions) gain improvement, and we speculate that the larger image-size can modelling the local context better.
>     2. DTFD try to split the whole bag into 3 sub-bags, but smaller bag size may result in larger label noise of sub-bags which may answer its performance drop.
>     3. The gap between LongMIL and TransMIL decreases given closer n and d. Full attention and LongMIL show small drops, since less interactions can be modelled with less patches.
>     4. LongMIL still out-performs full attention. We speculate that local attention also works better when dealing with the shape-varying WSI even with less n.
>
> * *BRACS, 224x224 VS 448x448*
> | Patch Encoder | UNI-224| | UNI-448| |
> |:---|:---|:---|:---|:---|
> | Method\Metric | F1 | AUC | F1 | AUC |
> | AB-MIL | 0.692±0.03 | 0.875±0.02 | 0.695±0.01 | 0.875±0.01 |
> | CLAM-SB | 0.640±0.06 | 0.844±0.03 | 0.654±0.03 | 0.851±0.02 |
> | DTFD-MIL | 0.655±0.03 | 0.878±0.02 | 0.625±0.03 | 0.839±0.01 |
> | TransMIL | 0.592±0.04 | 0.859±0.02 | 0.646±0.07 | 0.855±0.02 |
> | Full Attention | 0.715±0.04 | 0.884±0.02 | 0.700±0.04 | 0.874±0.02 |
> | LongMIL (ours) | 0.728±0.05 | 0.887±0.01 | 0.722±0.03 | 0.883±0.01 |
>
> >**W3 + Q3: Presentation: ...**
>
> We appreciate your advice and will definitely focus on refining the paper's presentation for the next version to enhance clarity.
>
> >**W4.2: ... consider using latest pathology foundation models ...**
>
> We have finished part of this experiments in this stage, which can be inferred in the general response.
>
> >**Q1: Since n >> d is MIL model-agnostic, would other approaches such as AB-MIL also suffer from low rank nature?**
>
> For AB-MIL, CLAM, etc., their attentions are in fact adaptive weighted average ($1 \times n$) over all patches, and there is no pair-wise interaction like self-attention does ($n \times n$ attention matrix).  It's hard to say the rank of a $1 \times n$ array, but we find that they are sparse like what doctors do: determine the lesion level of a slide based on some specific lesion regions. However, this process needs quite high semantics align with doctors. Conversely, our method with self-attention can be understood as trying to convert the feature representation (learned by patch-level self-supervised learning) aligning better to doctors' diagnosis level by introducing important context. After layers of attention, cls-token and average pooling on these features will do the same thing as AB-MIL.
>
> >**Q2: It seems LongMIL supersedes existing positional encodings (Or am I understanding this correctly?) - Can they be combined?**
>
> We have combined the rotary positional embedding, since we still need to determine the position information within the local areas (about a size of $20 \times 20$). We feel sorry to ignore this implementation detail and will add it into the next version.
>
>
> ### Ref:
>
> [1]. Dual-stream multiple instance learning network for whole slide image classification with self-supervised contrastive learning.
>
> [2]. Cross-scale multi-instance learning for pathological image diagnosis.

---

> > ### Comment · Reviewer_UesU · 2024-08-12
> > **Response**
> >
> > Thank you for addressing my concerns - I trust the authors in the revised version will do their best to address the clarity issue I raised. I increase my score accordingly.

---

> > > ### Author Response · Authors · 2024-08-13
> > >
> > > We appreciate the constructive feedback aimed at enhancing the clarity of our paper. We agree with the points and suggestions raised and will address them in the revised manuscript.

---

### Official Review · Reviewer_Ghwr · 2024-07-12

**Soundness:** 2
**Presentation:** 2
**Contribution:** 2
**Rating:** 3
**Confidence:** 5

**Summary:**

This paper focuses on the issue of attention computation for long sequences in WSI (Whole Slide Image) images. The authors first analyze how the low-rank nature of the long-sequence attention matrix constrains the representation ability of WSI modeling. They then propose a method using local attention masks to compute attention within local regions, followed by the computation of global attention. Experimental results demonstrate that the combination of local and global attention computations outperforms full attention.

**Strengths:**

1. The computation of attention for ultra-long sequences is a significant challenge in WSI slide-level feature learning, and addressing this issue is highly valuable.
2. The paper provides a detailed analysis of the low-rank and sparsity problems in the attention matrix of long sequences, based on which the use of local attention is proposed.
3. The authors propose the longMIL method, which achieves better results than the baseline in both subtyping and prognostic tasks.

**Weaknesses:**

1. Although the paper identifies and analyzes the bottleneck issues in long-sequence attention and proposes the use of local attention based on this analysis, there is a lack of innovation in using local attention. Other methods, such as LongViT, LongNet, and Prov-GigaPath, have also used local attention more elegantly. Additionally, the paper does not discuss the potential for directly transferring numerous attention optimization methods from the fields of computer vision (CV) and natural language processing (NLP).
2. The paper lacks a review of relevant literature, such as the aforementioned works.
3. In the experimental results, for instance, the classification results in Table 1 and the prognostic results in Table 2 show only a 0.01 improvement over full attention, which is not significant.
4. Although the paper claims to reduce computational costs, which is evident, it should provide corresponding comparisons to substantiate this claim.

**Questions:**

see weakness

---

> ### Author Rebuttal · Authors · 2024-08-06
>
> Dear Reviewer Ghwr,
>
> We appreciate your time and constructive feedback. We are glad that you found that our analysis and method are valuable. Below, please find our point-to-point response to your comments:
>
> > **W1 & W2: Although the paper identifies and analyzes the bottleneck issues in long-sequence attention and proposes the use of local attention based on this analysis, there is a lack of innovation in using local attention. Other methods, such as LongViT, LongNet, and Prov-GigaPath, have also used local attention more elegantly. Additionally, the paper does not discuss the potential for directly transferring numerous attention optimization methods from the fields of computer vision (CV) and natural language processing (NLP).
> The paper lacks a review of relevant literature, such as the aforementioned works.**
>
> We have great respect for the significant contributions made by Prov-GigaPath, LongViT, and LongNet, as well as the innovative dilated-attention method that underpins them. These advancements have brought substantial progress to the field of Computational Pathology. We regret that our initial submission did not adequately acknowledge these important works. The Prov-GigaPath paper, published in Nature on May 22, 2024, coincided with the NeurIPS submission deadline, while the LongViT was available as an arXiv preprint formatted as a technical report and titled as a 'Case Study' thus we missed it. We sincerely apologize for this oversight and have now included extensive discussions on these works in our rebuttal. We are committed to providing a more thorough analysis in the future version of our paper.
>
> For the detailed comparisons of our method and Prov-GigaPath, please infer to the general response, where we make systematical analysis including:
>
>     d_1. Method: their receptive field weigh more on x-axis than y-axis, however our method as 2D locality treat x-y equally. Please also check the figure illustration in the rebuttal PDF.
>     d_2. Contribution: we focus more on analyzing why previous transformers failed then deriving our method, while they empirically scale up to big data based on dilated attention.
>     d_3. We find that when their patch feature is not the best in some task cases, their heavily pretrained WSI head with problem in 'd_1' only shows sub-optimal performance.
>     We provide some quick experiments to compare their WSI-architecture and our method.
>
> >**W3: In the experimental results, for instance, the classification results in Table 1 and the prognostic results in Table 2 show only a 0.01 improvement over full attention, which is not significant.**
>
> It is not so easy to totally beat the full attention in performance since our main goal is to achieve comparable results as full attention but with less computational complexity. Even in the area of NLP currently, like LLM, the vanilla full attention is the most widely adopted choice [1][2][3] when equipped with the best hardware, although a lot of methods [4][5][6], and also the LongNet, are trying to replace it to solve long sequence problem. However, in the area of digital pathology given less computational resources, the inference speed is quite important. Moreover, we have tested the training and inference speed when dealing with larger resolution WSI or higher magnification like 40x with 0.25 mpp, the latency is unbearable even with flash attention. Whereas our method can highly alleviate the speed problem without performance drop (even improvement) if we need more detailed features in 40x.
>
>
> >**W4: Although the paper claims to reduce computational costs, which is evident, it should provide corresponding comparisons to substantiate this claim.**
>
> Please check A.6.5 and Figure 8 of our main paper manuscript, where we have tested the speed (time consumption in deployed GPU), and shown the comparisons in paper submission stage.
>
> The theoretical complexity is also included in the *line 232-236* of our main paper manuscript.
>
> If you have additional questions, we’re happy to engage further.
>
> ### Ref:
>
> *[1]. Touvron H, Martin L, Stone K, et al. Llama 2: Open foundation and fine-tuned chat models[J]. arXiv preprint arXiv:2307.09288, 2023.*
>
> *[2]. Jiang A Q, Sablayrolles A, Roux A, et al. Mixtral of experts[J]. arXiv preprint arXiv:2401.04088, 2024.*
>
> *[3]. Bai J, Bai S, Chu Y, et al. Qwen technical report[J]. arXiv preprint arXiv:2309.16609, 2023.*
>
> *[4]. Gu A, Dao T. Mamba: Linear-time sequence modeling with selective state spaces[J]. arXiv preprint arXiv:2312.00752, 2023.*
>
> *[5]. Sun Y, Dong L, Huang S, et al. Retentive network: A successor to transformer for large language models[J]. arXiv preprint arXiv:2307.08621, 2023.*
>
> *[6]. Peng B, Alcaide E, Anthony Q, et al. Rwkv: Reinventing rnns for the transformer era[J]. arXiv preprint arXiv:2305.13048, 2023.*

---

> > ### Comment · Reviewer_Ghwr · 2024-08-09
> > **response**
> >
> > Since the two-stage method is adopted in this paper, the first stage has used the pre-trained model to extract features offline. The second stage feature dimension has been smaller. Usually the number of effective patches (256) for a WSI image is between a few thousand and ten thousand, and the second stage network is smaller. In the experiment, the second stage of the network usually only needs a few simple transformer layers, which can be completed in a few minutes or ten minutes of training. In view of this, if the local attention that cannot be proposed in this paper is not to solve the performance bottleneck and other effects of full attention caused by too many tokens, it cannot exceed full attention (I think too many tokens will lead to inefficient learning, full attention is not necessarily better than local attention performance), and I have doubts about the actual value in terms of solving computational efficiency.

---

> > > ### Author Response · Authors · 2024-08-10
> > >
> > > > the number of effective patches (256) for a WSI image is between a few thousand and ten thousand.
> > >
> > > This is not always the case. There are various cases facing larger patch numbers need computational efficiency:
> > >
> > >     1. 40x magnification, with about n=40k~70k patches (also shown in Fig. 8), we have test training speed of full attention, which is about 25x to 35x times than ours. We have talked about this in the caption of Fig. 8, and also show some experimental results of performance in Table 5 of A.6.4. Though currently 40x is not the mainstream, we speculate this is caused by 20x captures better context thus performs better using AB-MIL (without context modelling ability) paradigm. Moreover, there is study [1] show better performance via 40x.
> > >     2. Overlapped patching, it will be resulting 2~4x patches if the overlap ratio is 0.25~0.5. This helps alleviating the edge effect of image modelling neither by CNN nor ViT.
> > >     3. Some slides contain >=2 histology tissues; Survival prediction need multiple slides for one patience.
> > >
> > > > if the local attention that cannot be proposed in this paper is not to solve the performance bottleneck and other effects of full attention caused by too many tokens... I have doubts about the actual value in terms of solving computational efficiency.
> > >
> > > We find that you are doubting on the motivation of the whole paper, but our paper shares the same motivation to the structure of TransMIL, HIPT and even Prov-Gigapath (to the best of our knowledge, in Prov-Gigapath, they also use patch size 256 and extract patch features from stage-1, then perform stage-2 slide-level 'dilated attention encoder' fine-tuning ). An interesting point is that there are very limited papers using full attention, even in the famous works like UNI and Prov-Gigapath, which we believe, is incurred by its unacceptable complexity.
> > > The complexity has impeded it being widely use or scaling to larger data setting, including:
> > >
> > >     1. 40x, overlapped patching ... as we mentioned above.
> > >     2. pretraining on over 100k slides just like Prov-Gigapath do.
> > >     3. slow speed in the clinical or deployed setting, where the GPU hardware is not so good as training.
> > >     4. using larger feature embedding d (e.g. carrying both high-level and low-level feature) as we mentioned in the line 207~208 of our paper: 'An intuitive modification to handle the low-rank problem is to set a larger embedding size d, but this makes computational complexity O(n^2 d) more severe...'
> > >     5. more transformer layers, as you said.
> > >
> > > These prospective directions or future work which may improve the WSI diagnosis or prognosis, we think, can be better implemented via our method. Though is this paper we cannot include all above topics, we are actively working to enhance these factors so that the model becomes a more valuable resource for the community.
> > >
> > > ### Ref:
> > > [1]. Yu, Jin-Gang, et al. "Prototypical multiple instance learning for predicting lymph node metastasis of breast cancer from whole-slide pathological images." Medical Image Analysis 85 (2023): 102748.

---

> > > > ### Comment · Reviewer_Ghwr · 2024-08-13
> > > >
> > > > Since the author claims that other work is also addressing local attention, what is the essential difference between local attention in this paper? There are so many local attention innovations in CV and NLP that can be directly applied to pathological images. The number of tokens in the NLP space is much larger than the tens of thousands of tokens here. I don't think it's a neurips level of innovation enough. I respect the opinion of other reviewers, but my final rating is still negative.

---

> > > > > ### Comment · Reviewer_Ghwr · 2024-08-13
> > > > >
> > > > > I revised my rating.

---

> > > > > > ### Comment · Reviewer_UUFY · 2024-08-14
> > > > > >
> > > > > > In response to Ghwr:
> > > > > >
> > > > > > I agree that there are many works that focus on efficient / local attention methods, but disagree that these works diminish the contributions of this work. In my view, this work addresses a very nuanced-but-interesting/relevant problem in Transformer architectures for WSIs - namely, how to interpolate positional embeddings to gigapixel imagery + avoiding rank collapse. Comparisons with local attention (HIPT) and efficient attention (TransMIL, GigaPath) are performed, with LongMIL doing better.
> > > > > >
> > > > > > I believe the proposed tricks to Transformer Attention implementation for WSIs would be insightful for researchers working in pathology. I agree that similar works have addressed this issue (e.g., GigaPath), but think the experimentation is different enough and having >1 article investigating this topic is overall useful to have, which is why I have rated this as Borderline / Weak Accept (leaning closer to Weak Accept).

---

### Official Review · Reviewer_GgQd · 2024-07-16

**Soundness:** 3
**Presentation:** 4
**Contribution:** 4
**Rating:** 7
**Confidence:** 5

**Summary:**

The authors point out that that MIL often has insufficient ability to offer accurate slide level classifications.  There is a long (now) history of attempting to better consider sub-slide level context in the aggregation function.  TransMIL, GNN-based methods all have provided attempts to this end.

The authors argue that transformer based aggregatioins functions have limited ability to consider both local and global context for tile-attention-rankings.  They provide theoretical arguments for why a "low-rank bottleneck" exists because the total number of patch embeddings is much greater than the embedding size.

The method addresses this limitation but calculating local self attention followed by pooling function before doing a "global" self attention function.

In main paper experiments are done with BRACS dataset using two different encoders for feature extraction.  The proposed method is benchmarked against cotext aware and non-context aware aggregations functions.  Similar experiment is done with survival predictions.  Extensive additional supporting experiments are in supplement.

**Strengths:**

The paper is well written and thorough.  The authors are very meticulous about addressing the problem proposed with traditional transformer based aggregators.  The claims are backed by the results.  The supplement provides extensive figures and ablation studies as well as additional details (including memory efficiency studies) on the methodologies.

**Weaknesses:**

A recent preprint (https://arxiv.org/abs/2407.07841) shows that context aware aggregations functions offer less performance boost over ABMIL when you have a high feature extraction encoder.  In this work the encoders used are less robust than the now publically available encoder (UNI, Gigapath and Virchow). All of these are very recently released so is understanable that they are not part of this submission.
 Going forward these encoders should be used for any aggregation function assessment.   The supplemental table showing that the boost of this method is much stronger with an image net pretrained cnn.

**Questions:**

In final version of paper, can you please improve orientation of figure 3.  It is hard to zoom in sufficiently to understand second panel.  It would also benefit from having sub labels (eg. a, b, c) to improve the legend description of the sub panels.

**Limitations:**

I concur with listed limitations.  I have pointed out other limitations in the weakness section.

---

> ### Author Rebuttal · Authors · 2024-08-06
>
> Dear Reviewer GgQd,
>
> We would like to express our sincere gratitude for your thoughtful review and insightful feedback on our manuscript. We appreciate your recognition of the thoroughness and approach taken in addressing the limitations of traditional transformer-based aggregators. Your positive evaluation of our work's soundness, presentation, and contribution is greatly encouraging.
>
> Below, we provide responses to your comments and questions:
>
> >**W1: A recent preprint (https://arxiv.org/abs/2407.07841) shows that context aware aggregations functions offer less performance boost over ABMIL when you have a high feature extraction encoder. In this work the encoders used are less robust than the now publicly available encoder (UNI, Gigapath and Virchow). All of these are very recently released so is understandable that they are not part of this submission. Going forward these encoders should be used for any aggregation function assessment. The supplemental table showing that the boost of this method is much stronger with an image net pretrained cnn.**
>
> We find this question being very constructive and insightful to the completeness our paper.
> On the one hand, we have performed some experiments on stronger patch encoder including UNI and GigaPath in this stage (check it in the general response), with main findings:
>
>     1. Our method continues to outperform previous work in complicated tasks such as BRACS and survival prediction, demonstrating a notable performance boost in consistency compared to other methods, with the exception of TransMIL.
>     2. The results in the TCGA-BRCA tumor subtyping task are similar across almost all the methods. We speculate that this task may have reached an upper limit when using a strong patch encoder. In future versions, we plan to validate our method with robust encoders on larger datasets to further explore the potential for improvement.
>
> On the other hand, motivated by this suggestion, we realize that strong patch encoders with high-level semantics may lose important low-level spatial-context or fine-grained details. Thus, in future work we may be going to aggregate more features from different depth of layers in ViT or different pretrained ViT. This may not only include more detailed spatial contexts, but also helps improving the feature size $d$.
>
> >**Q1: In final version of paper, can you please improve orientation of figure 3. It is hard to zoom in sufficiently to understand second panel. It would also benefit from having sub labels (eg. a, b, c) to improve the legend description of the sub panels.**
>
> Thank you for your detailed feedback on Figure 3. We understand that the current orientation and labeling make it difficult to interpret, particularly the second panel. To address this issue, we will take actions to improve the clarity and readability of the figure in the final version of our manuscript.

---

### Official Review · Reviewer_UUFY · 2024-07-16

**Soundness:** 3
**Presentation:** 2
**Contribution:** 3
**Rating:** 6
**Confidence:** 5

**Summary:**

This work examines the problem of extrapolating Transformer attention to long sequences in WSI representation learning. The main technical contribution is in examining the low-rank bottleneck problem of Transformer attention for WSIs, and proposing LongMIL which introduces modifications via local attention masking + 2D ALIBI in order to improve the rank of the attention matrix and enable extrapolation capabilities.

**Strengths:**

- Core contribution of this work is novel and would have a lot of interest in the CPath community. Extrapolating to long contexts is an exciting problem that has had little investigation (outside of Prov-GigaPath). I believe this work can be re-organized in presenting a more systematic understanding of they key components needed to extrapolating to long contexts.
- Supplement includes some interesting ideas and ablation experiments. A.4 discusses similarities and differences between HIPT, adding local mask attention, and 2D alibi. A.6.1 includes a comparison with the ViT-S in HIPT for equivalent comparisons. A.6.3 examines different hyper-parameters during Transformer training. A.6.4 looks at the difference between 20X and 40X magnification. A.6.6. ablates other straightforward extensions of Transformer attention with subquadratic complexity (including Mamba and V-Mamba).
- Figures are illustrative (in both main text and Supplement).

**Weaknesses:**

- Main limitation of this work is that a comparison with Prov-GigaPath [1] (a concurrent work that appeared at the time of NeurIPS submission) is warranted. Prov-GigaPath also presents overlapping contributions in solving this problem, though I think there is room for more than 1 study examining this problem.
- The writing feels a bit rushed and informal. I was often scrolling back and forth to understand the different comparisons being made. Ideally, there should be one table for results that compare MIL architectures and one table for ablating pretrained encoders. Other areas where the writing / presentation of figures and results could be significantly polished:
- - "We omit the ResNet-50 embedding for survival prediction since it get quite low and unacceptable results." Informal and non-scientific.
- - Section 4.3 should be able to summarize the findings in the Supplement in a more clear and descriptive manner.
- - One of the main issues I see in this work is that the authors are juggling many different encoders for different tasks. A.6.1 compares ViT-S Lunit and HIPT for TCGA-BRCA subtyping (evaluating multiple MIL models across multiple encoders on 1 task), but A.6.2. shows Resnet-50 features for BRACS and TCGA-BRCA subtyping (evaluating multiple MIL models across multiple tasks with the same encoder). Many of these issues can be drastically simplified if the authors were to use an encoder not pretrained on TCGA such as Prov-GigaPath [1], UNI [2], or PLIP [3] - with the main emphasis on comparing LongMIL with other competing works. Ref [4] provides an example on how the findings of this work can be better organized.

References
1. Xu, H., Usuyama, N., Bagga, J., Zhang, S., Rao, R., Naumann, T., Wong, C., Gero, Z., González, J., Gu, Y. and Xu, Y., 2024. A whole-slide foundation model for digital pathology from real-world data. Nature, pp.1-8.
2. Chen, R.J., Ding, T., Lu, M.Y., Williamson, D.F., Jaume, G., Song, A.H., Chen, B., Zhang, A., Shao, D., Shaban, M. and Williams, M., 2024. Towards a general-purpose foundation model for computational pathology. Nature Medicine, 30(3), pp.850-862.
3. Huang, Z., Bianchi, F., Yuksekgonul, M., Montine, T.J. and Zou, J., 2023. A visual–language foundation model for pathology image analysis using medical twitter. Nature medicine, 29(9), pp.2307-2316.
4. Park, N. and Kim, S., How Do Vision Transformers Work?. In International Conference on Learning Representations.

**Questions:**

Would the authors be able to address my concern in updating the results of this work to using a different encoder (to simplify the presentation of results)? Choice of pretrained encoder should not matter significantly (as the main focus is in fairly comparing LongMIL), but having 1-2 comparisons would be nice in a Supplemental Figure.

Rating at the time of reviewing this work is slightly negative, but am enthusiastic of this work and would raise my rating to borderline / weak accept if my concerns were addressed.

---

> ### Author Rebuttal · Authors · 2024-08-06
>
> Dear Reviewer UUFY,
>
> We appreciate your time and valuable feedback. We are glad that you found the formulation and analysis being novel and sound, and the figures and ablations being illustrative. Below, please find our point-to-point response to your comments:
> > **W1: Main limitation of this work is that a comparison with Prov-GigaPath (a concurrent work that appeared at the time of NeurIPS submission) is warranted. Prov-GigaPath also presents overlapping contributions in solving this problem, though I think there is room for more than 1 study examining this problem.**
>
> Since another reviewer Ghwr also points out this problem, we post the detailed comparisons in the general response, where we make systematical comparisons to Prov-GigaPath including:
>
>     d_1. Method: their receptive field weigh more on x-axis than y-axis, however our method as 2D locality treat x-y equally.
>     d_2. Contribution: we focus more on analyzing why previous transformers failed then deriving our method, while they empirically scale up to big data based on dilated attention.
>     d_3. We find that when their patch feature is not the best in some task cases, their heavily pretrained WSI head with problem in 'd_1' only shows sub-optimal performance.
>     We provide some quick experiments to compare their WSI-architecture and our method.
>
> >**W2: The writing feels a bit rushed and informal. I was often scrolling back and forth to understand the different comparisons being made. Ideally, there should be one table for results that compare MIL architectures and one table for ablating pretrained encoders. Other areas where the writing / presentation of figures and results could be significantly polished ...**
>
> We thank a lot for your suggestions on the presentation of our paper, we will polish it for better reader-experience in next version.
>
> >**W2.3 + Q1: Would the authors be able to address my concern in updating the results of this work to using a different encoder (to simplify the presentation of results)? Choice of pretrained encoder should not matter significantly (as the main focus is in fairly comparing LongMIL) but having 1-2 comparisons would be nice in a Supplemental Figure.**
>
> We indeed find that there are more and more stronger patch encoders pretrained without TCGA should be validated, which is also highlighted by almost all the other reviewers. So, we also add it into the general response, including both UNI and GigaPath pretrained patch encoders with main findings:
>
>     1. Our method still outperforms previous work for some complicated tasks like BRACS and survival prediction.
>     2. The results in TCGA-BRCA tumor-subtyping are similar for almost all the WSI methods. We speculate that this task as binary classification might have reached some sort of upper limit when equipped with strong patch encoder. We will be going to evaluate our method on more difficult tasks in future version.

---

> > ### Author Response · Authors · 2024-08-13
> >
> > Thank you again for your review.
> > Considering that it is the last day of the discussion period, we would like to confirm whether our rebuttal has adequately addressed the concerns you raised.
> >
> > We continue to welcome any supplementary observations or clarification to bolster our work.

---

> > > ### Comment · Reviewer_UUFY · 2024-08-14
> > >
> > > Dear authors,
> > >
> > > Thank you for your response. I am happy with the author's rebuttal, and have thus raised my score.

---

> > > > ### Author Response · Authors · 2024-08-14
> > > >
> > > > Thank you very much for the acknowledgement. We deeply appreciate your time and effort in the review. We will further revise manuscript in next version based on your suggestions.

---

### Author Rebuttal · Authors · 2024-08-06

Thanks to all the reviewers for your time and effort during the review process. We appreciate that you found our work insightful and solid.

We have responded to each reviewer individually, uploaded a rebuttal PDF, and collected the below response to general concerns. If you find our answers responsive to your concerns, we would be grateful if you considered increasing your score, and if you have additional questions, we’re happy to engage further.


> **All reviewers suggest evaluating on better pre-trained patch encoders**

Here, we list experiment results of tumor-subtyping on BRACS and TCGA-BRCA.
We find that our method still outperforms previous method in BRACS with UNI and GigaPath. But for TCGA-BRCA tumor-subtyping, almost all the methods show similar results, which may because the binary classification is too simple, and this task meet its upper-bound given so strong patch feature encoders. We will experiment on more data with higher complexity in future work.

* *BRACS, tumor subtyping*
| Patch Encoder | UNI | UNI | GigaPath | GigaPath |
|:---|:---|:---|:---|:---|
| Slide Method\Metric | F1 | AUC | F1 | AUC |
| AB-MIL | 0.692±0.033 | 0.875±0.020 | 0.640±0.022 | 0.837±0.010 |
| CLAM-SB | 0.640±0.057 | 0.844±0.025 | 0.624±0.023 | 0.826±0.014 |
| DTFD-MIL | 0.655±0.031 | 0.878±0.022 | 0.610±0.032 | 0.843±0.017 |
| TransMIL | 0.592±0.036 | 0.859±0.023 | 0.599±0.058 | 0.838±0.048 |
| Full Attention | 0.715±0.043 | 0.884±0.017 | 0.663±0.023 | 0.850±0.018 |
| LongMIL (ours) | 0.728±0.045 | 0.887±0.008 | 0.673±0.023 | 0.856±0.015 |

* *TCGA-BRCA, tumor subtyping*
| Patch Encoder | UNI | UNI | GigaPath | GigaPath |
|:---|:---|:---|:---|:---|
| Slide Method\Metric | F1 | AUC | F1 | AUC |
| AB-MIL | 0.865±0.039 | 0.945±0.018 | 0.872±0.038 | 0.946±0.021 |
| CLAM-SB | 0.862±0.031 | 0.943±0.020 | 0.864±0.049 | 0.937±0.027 |
| DTFD-MIL | 0.867±0.034 | 0.941±0.024 | 0.870±0.035 | 0.937±0.034 |
| TransMIL | 0.853±0.049 | 0.949±0.019 | 0.830±0.048 | 0.934±0.020 |
| Full Attention | 0.849±0.043 | 0.942±0.017 | 0.860±0.041 | 0.946±0.023 |
| LongMIL (ours) | 0.863±0.033 | 0.945±0.008 | 0.871±0.030 | 0.947±0.022 |

Due to the limited time and large model architecture of UNI and GigaPatch, we only experiment survival prediction on TCGA-BRCA since its features are extracted in tumor-subtyping.

* *TCGA-BRCA, survival prediction*
| Patch Encoder | UNI | GigaPath |
|:---|:---|:---|
| Slide Method\Metric | c-index | c-index|
| AB-MIL | 0.630±0.054 | 0.635±0.033 |
| AMISL | 0.627±0.080 | 0.620±0.040 |
| DS-MIL | 0.616±0.034 | 0.612±0.086 |
| TransMIL | 0.598±0.059 | 0.599±0.064 |
| Full Attention | 0.638±0.056 | 0.617±0.069 |
| LongMIL (ours) | 0.656±0.061 | 0.645±0.055 |


> **Reviewers UUFY and Ghwr concern on the comparison to Prov-GigaPath / LongViT WSI-Architecture**

Although reviewers UUFY and Ghwr point out that both our paper and Prov-GigaPath use similar local attention mechanism for efficient transformer modelling in slide-level, we find that there are some important differences between them.

    1. The motivation /contribution: our paper not only focus on proposing an efficient self-attention mechanism for WSI, but also showing analysis on why some previous work like Roformer and TransMIL fail for WSI from the low-rank perspective, which we believe to be insightful to the digital pathology community. However, both the Prov-GigaPath and LongViT focus on scaling up to a large-scale of data with pre-training, which is more empirical. We believe that our analysis may also work for Prov-GigaPath and could be one potential explain on why Prov-GigaPath success and how to improve further.
    2. The method details: Prov-GigaPath does not treat interactions inside x-axis and y-axis equally, though the 2-d positional embedding is applied. By putting all patches into a 1-d sequence in a 'z-scan' manner like ViT, their 1-d local attention focus more on x-axis but less on y-axis, as depicted in Fig. 1 of our rebuttal PDF. Although this can be alleviated by their higher-level dilated attention term, the x-y inequality still exists. Whereas, our local-attention is designed for 2-d (based on 2d Euclid distance), thus treat them equally.
    3. The pretrained Prov-GigaPath WSI-head seems relying heavily on their own patch-pretrained encoder, which may be a potential barrier to wide usage, e.g. there are still some cases when GigaPath patch features weaker than UNI or Conch, as posted in the github repo of UNI. The WSI pretraining is indeed useful as the key to their superior performance, which covers their problem of spatial inequality on x and y. When dealing with the case 'BRACS', as shown in the following table, our method (even AB-MIL) with better UNI feature can outperform their 'worse patch feature with stronger pretrained slide encoder'.

Experimentally, we here perform evaluation on the two WSI-level architectures. Since the WSI params are pre-trained in Prov-GigaPath, we also experiment it using random initialization for fair comparison. For the mismatch of UNI patch encoder and GigaPath WSI head, we add a nn.Linear layer as a feature projector. We find that the pre-training plays a key role to the success of Prov-GigaPath WSI-head, since transformers are much more over-parameterized than previous simple attention-based MIL. Given the limited time, we are going to validate on more tasks in future version.

* *LongMIL V.S. Prov-GigaPath in slide-level on BRACS, tumor subtyping*
| Patch Encoder | UNI | UNI | GigaPath | GigaPath |
|:---|:---|:---|:---|:---|
| Slide Method\Metric | F1 | AUC | F1 | AUC |
| AB-MIL | 0.692±0.033 | 0.875±0.020 | 0.640±0.022 | 0.837±0.010 |
| TransMIL | 0.592±0.036 | 0.859±0.023 | 0.599±0.058 | 0.838±0.048 |
| GigaPath (random init) | 0.648±0.041 | 0.837±0.033 | 0.627±0.038 | 0.808±0.038 |
| GigaPath (pre-trained) | 0.668±0.026 | 0.861±0.030 | 0.677±0.033 | 0.862±0.034 |
| LongMIL (ours) | 0.728±0.045 | 0.887±0.008 | 0.673±0.023 | 0.856±0.015 |

---

### Decision · Program_Chairs · 2024-09-25

**Decision:**

Accept (poster)

**Comment:**

This submission investigating transformers for histopathology image analysis received mixed review comments. The rebuttal added more comparison results. Reviewers raised scores or reinforced positive score after reading rebuttal and discussion with authors. The key discussing point of this submission is the contribution of its methodological innovations. The AC heard majority of the reviewers recognizing the paper's domain-oriented contribution. Despite the attention mechanism has been used in the literature, this paper has made solid method on designing the transformer architecture such that it fits the practical challenge for SWI data processing. Overall, the strengths of the work such as rigor & extensive experimentation outweigh its weaknesses. A final decision of acceptance is recommended.